# The ratio of transverse to longitudinal turbulent velocity statistics for aircraft measurements

Jakub L. Nowak[1], Marie Lothon[2], Donald H. Lenschow[3], and Szymon P. Malinowski[1]

[1]Institute of Geophysics, Faculty of Physics, University of Warsaw, Warsaw, Poland
[2]Laboratoire d'Aérologie, University of Toulouse, CNRS, UPS, Toulouse, France
[3]National Science Foundation National Center for Atmospheric Research, Boulder, CO, USA

**Correspondence:** Jakub L. Nowak (jakub.nowak@fuw.edu.pl)

**Abstract.** The classical theory of homogeneous isotropic turbulence predicts the ratio of transverse to longitudinal structure functions or power spectra equal to 4/3 in the inertial subrange. For the typical turbulence cascade in the inertial subrange, it also predicts a power law scaling with an exponent of +2/3 and -5/3 for the structure functions and the power spectra, respectively. The goal of this study is to document the statistics of those ratios and exponents derived from aircraft observations, quantify their departures from theoretical predictions and point out the differences among the aircraft.

We estimate the transverse-to-longitudinal ratios and the scaling exponents from in-situ high-rate turbulence measurements collected by three research aircraft during four field experiments in two regimes of the marine atmospheric boundary layer: shallow trade-wind convection and subtropical stratocumulus. The bulk values representing the inertial subrange were derived by fitting power law formulas to the structure functions and power spectra computed separately for the three components of the turbulent wind velocity measured in horizontal flight segments. The composite scale-by-scale transverse-to-longitudinal ratios were derived by averaging over the segments at common non-dimensional scales.

The variability in the results can be attributed to how the wind velocity components are measured on each aircraft. The differences related to environmental conditions, e.g. between characteristic levels and regimes of the boundary layer, are of secondary importance. Experiment-averaged transverse-to-longitudinal ratios are 23-45 % smaller than predicted by the theory. The deviations of average scaling exponents with respect to the theoretical values range from -34 to +47 % for structure functions and from -24 to +22 % for power spectra, depending on experiment and velocity component. The composite scale-by-scale transverse-to-longitudinal ratios decrease and increasingly depart from 4/3 with decreasing scale, in contrast to previous experimental studies on local isotropy. The reason for the disagreement in transverse-to-longitudinal ratios between the observations and the theory remains uncertain.

## 1 Introduction

### 1.1 Theoretical background

According to the theory of homogeneous isotropic turbulence formulated by Kolmogorov (Kolmogorov, 1941), which is introduced in many classical textbooks (e.g. Pope, 2000, ch. 6), the second-order longitudinal and transverse velocity structure

functions in the inertial subrange can be approximated as

$$D_L(r) = B_L(\epsilon r)^{2/3}, \quad D_T(r) = B_T(\epsilon r)^{2/3}, \tag{1}$$

respectively, where $r$ is the separation distance, $\epsilon$ is the turbulence kinetic energy dissipation rate and $B_L$, $B_T$ are constants. Due to isotropy and homogeneity, the ratio of those structure functions is

$$\frac{D_T}{D_L} = \frac{4}{3}. \tag{2}$$

Analogous to structure functions, one-dimensional longitudinal and transverse velocity power spectra in the inertial subrange are

$$P_L(k) = C_L \epsilon^{2/3} k^{-5/3}, \quad P_T(k) = C_T \epsilon^{2/3} k^{-5/3}, \tag{3}$$

respectively, where $k$ is the longitudinal wavenumber, $C_L$, $C_T$ are constants and

$$\frac{P_T}{P_L} = \frac{4}{3}. \tag{4}$$

Only one of the four constants needs to be determined experimentally, as due to isotropy they are functionally related. The approximate values are $B_L \approx 2.0$, $B_T \approx 2.6$, $C_L \approx 0.49$, $C_T \approx 0.65$ (e.g. Saddoughi and Veeravalli, 1994; Sreenivasan, 1995).

Kolmogorov did not specify precise limits for the applicability of his theory. Instead, his famous hypotheses state that sufficiently far from the boundaries of the domain for a turbulent fluid (e.g. the surface and top of the atmospheric boundary layer (ABL)) and for sufficiently large Reynolds number, there exist a range of scales where the turbulent velocity statistics are isotropic and universal. Nevertheless, the simplicity of this theory is considered as advantageous in experimental practice. On the other hand, we note that there have been some recent theoretical advancements examining non-stationary, non-homogeneous or non-isotropic conditions (e.g. Gomes-Fernandes et al., 2015; Wacławczyk et al., 2022).

Longitudinal direction is defined by the 2-point separation vector $\boldsymbol{r}$. The directions perpendicular to it are transverse (c.f. Pope, 2000, ch. 6.2). In experimental works, typically frozen turbulence approximation is invoked to compute multi-point statistics, such as structure functions or power spectra. Then, longitudinal direction is determined by the velocity vector of a probe with respect to turbulent medium (c.f. Pope, 2000, ch. 6.5). In the case of rapidly moving platform, e.g. aircraft, taking the limit of infinite probe velocity allows to consider measurement record as an instantaneous state of turbulent medium, e.g. air, along a 1-dimensional segment. In the case of a probe which is stationary with respect to ground, e.g. at a meteorological mast, the velocity of the probe with respect to air is simply opposite to the air velocity with respect to ground, i.e. wind. Classical Taylor's hypothesis (Taylor, 1938) allows to use mean wind velocity to convert measured timeseries into an instantaneous state of turbulent air along a 1-dimensional segment oriented with the mean wind. Such approximations are justified as long as the probe velocity - true air speed for aircraft or mean wind for a mast - is much larger than turbulence velocity scale. Often, measurement method implies that the longitudinal direction is in horizontal. Then, one of the transverse directions is vertical and the remaining third dimension is called lateral. Following a typical convention, we denote the longitudinal, lateral and vertical velocity components as $u$, $v$ and $w$, respectively. As both $v$ and $w$ are transverse, $D_v/D_u$ and $P_v/P_u$ as well as $D_w/D_u$ and $P_w/P_u$ are expected to equal 4/3 in homogeneous isotropic turbulence.

## 1.2 Measurements in the surface layer

The local isotropy hypothesis has been extensively tested in wind tunnels (e.g. Saddoughi and Veeravalli, 1994) and with ground-based measurements in the atmospheric surface layer (e.g. Kaimal et al., 1972; Katul et al., 1995, 1997; Siebert and Muschinski, 2001; Chamecki and Dias, 2004). The ground-based experiments typically rely on 3-component ultrasonic anemometers mounted at various heights $z$ above the surface.

Kaimal et al. (1972) analyzed measurements collected at three heights on a 32 m tower in a range of stable and unstable conditions during Kansas 1968 experiment (Haugen et al., 1971). They examined the onset of local isotropy using scale-by-scale ratios of power spectra and found that the isotropic value of 4/3 is gradually approached with decreasing scale. For $P_w/P_u$, this is observed at wavelengths $\lambda$ comparable to $z$ in unstable conditions and a tenth of Obukhov length $L_O$ in stable conditions. In general, the critical wavelength decreases with stability parameter $z/L_O$. $P_v/P_u$ reaches the isotropic ratio at scales about 8 times larger than $P_w/P_u$. The onset of local isotropy is directly related to the onset of universal Kolmogorov scaling in $P_w$. $P_u$ exhibits the -5/3 scaling as the first, i.e. at largest scales, while the vertical as the last, i.e. at smallest scales. Therefore, the adequate scaling observed in $P_w$ implies local isotropy. Kaimal et al. (1972) explained their results by the combined effects of shear and buoyancy on small-scale eddies. They argued that only eddies with timescales small compared with the production scales can be isotropic.

Katul et al. (1995) and Katul et al. (1997) performed similar measurements up to $z \approx 5$ m in unstable surface layer. They found that $D_v/D_u$, $D_w/D_u$ are approximately 4/3 for the scales below $z/2$. Katul et al. (1995) suggested that the two mechanisms responsible for anisotropy - buoyancy and wind shear - superimpose under stable conditions, where buoyancy damps vertical while shear enhances horizontal fluctuations, but counteract under unstable conditions, where buoyancy strengthens vertical fluctuations instead. Therefore, local isotropy can be more easily achieved in unstable but is not observed down to the very small scales in stable surface layers as previously shown by Kaimal et al. (1972) before.

Siebert and Muschinski (2001) tested their ultrasonic anemometer at $z =$2.8 and 5.5 m. They obtained the ratios $P_v/P_u$ and $P_w/P_u$ approaching 4/3 at $\lambda < z/2$ in agreement with Kaimal et al. (1972). It was noted that $P_w/P_u$ decreases at small scales because the spectral transfer function representing the low-pass filtering at scales close to sonic path drops more rapidly with frequency for $w$ than for $u$.

Chamecki and Dias (2004) measured wind velocity fluctuations at $z \approx 4$ m in a range of stable and unstable conditions. They found that although $D_v/D_u$ and $P_v/P_u$ reach 4/3 at small scales, $D_w/D_u$ and $P_w/P_u$ are systematically smaller than 4/3 down to the scale of instrument resolution (conservatively estimated as 36 cm for horizontal and 63 cm for vertical components). $D_w/D_u$ was noticed to be even further from the isotropic value then $P_w/P_u$. It was attributed to a relatively shorter extension of inertial range in structure functions in comparison to power spectra.

## 1.3 Measurements above the surface layer

The measurements of turbulent wind velocity far from the surface require more advanced platforms. For example, Kaimal et al. (1976) probed the convective mixed layer up to ~1200 m with a tethered kite balloon during the Minnesota 1973

experiment (Readings et al., 1974). Their system involved 5 lightweight cup anemometers mounted along the rope. Although they do not explicitly discuss the local isotropy, their results imply the spectral ratios $P_v/P_u$ and $P_w/P_u$ are 4/3 in the mixed layer at the scales smaller than a tenth of the ABL height $z_i$. Kaimal et al. (1982) reached the lower mixed layer with the Boulder Atmospheric Observatory 300 m tower which was instrumented with sonic anemometers analogous to those used in the surface layer (see sec. 1.2). They observed isotropic value for $P_v/P_u$ and $P_w/P_u$ at $\lambda < z/2$. Siebert et al. (2006b) analyzed two measurement series collected in shallow cumulus clouds at ∼760 m and ∼1540 m with an instrumented platform ACTOS (Siebert et al., 2006a), including a sonic anemometer, carried by a tethered balloon. In their first experiment, $P_v/P_u$ and $P_w/P_u$ were approximately 4/3 in the range of scales about 0.4-8 m. In the second experiment, those ratios were also relatively close to the isotropic value, however with $P_v/P_u$ systematically higher and $P_w/P_u$ systematically lower than 4/3.

Nowak et al. (2021) analyzed measurements from the same platform ACTOS, however carried by a helicopter, in coupled and decoupled marine stratocumulus-topped ABLs during ACORES campaign (Siebert et al., 2021). They compared turbulence kinetic energy dissipation rates $\epsilon$ derived separately from $u$ and $w$ by fitting the Kolmogorov scaling equations Eqs. (1) and (3) to structure functions and power spectra in the range 0.4–40 m which was assumed to represent the inertial subrange. The ratios $\epsilon_w/\epsilon_u$ used in that work are in fact equivalent to $(3D_w/4D_u)^{3/2}$ or $(3P_w/4P_u)^{3/2}$ within the selected range of scales. Those derived from power spectra were systematically lower than derived from structure functions (the reason thereof was not clear). The results exhibits strong local fluctuations which was attributed to the steep helicopter ascents or descents. The averaged values are nearly constant across the depth of the coupled ABL (equivalent to $D_w/D_u \approx 1.16$ and $P_w/P_u \approx 0.88$). In the decoupled ABL, they were smaller and differed between its sublayers (equivalent to $D_w/D_u \approx 0.74$, $P_w/P_u \approx 0.59$ in the lower part and $D_w/D_u \approx 0.53$, $P_w/P_u \approx 0.47$ in the cloud). Such a variation was explained by the separation of the ABL into two major circulations featuring contrasting turbulence properties.

In the same study, Nowak et al. (2021) presented scale-by-scale spectral ratio $P_w/P_u$ computed from horizontal flight segments at a few heights. In the coupled ABL (at three levels therein), the isotropic value 4/3 was approximately attained for the scales 5–100 m. The ratio decreases for larger scales presumably due to the finite distance from the surface or the capping inversion ($z_i \sim 850$ m) and for smaller scales arguably due to instrumental issues. In the decoupled ABL, the maximum $P_w/P_u$ was larger than 4/3 at intermediate scales. The range where $P_w/P_u \gtrsim 4/3$ was narrower and differed between the four considered levels. Interestingly, it was related to the fact that the depths of the two sections of the decoupled ABL are shallower than the coupled ABL although the entire decoupled one is deeper in total ($z_i \sim 1050$ m). It was also speculated that the scales where $P_w/P_u > 4/3$ might represent the typical sizes of surface layer plumes for the lowest segment and cloud-top downdrafts for the highest segment.

Despite several studies reviewed above which exploited very tall tower observatories, unique tethered balloon or helicopter-borne platforms, turbulent wind velocity far from the surface is typically measured in-situ from research aircraft in the course of intensive field experiments (e.g. Nicholls, 1984; Duynkerke et al., 1995; Lenschow et al., 2000; Malinowski et al., 2013; Brilouet et al., 2021). Research aircraft capable of turbulence measurements are often equipped with a five-hole radome probe with pressure transducers and a Pitot tube for air velocity measurements, and an inertial navigation system coupled to a GPS unit. The three components of the wind velocity are computed by adding the aircraft velocity with respect to the earth and the

velocity of air with respect to the aircraft which is inferred from true air speed (TAS), and attack and sideslip angles (Brown et al., 1983; Lenschow, 1986; Lenschow and Spyers-Duran, 1989). TAS is obtained from the measurements of total and static pressure whereas attack and sideslip angles are determined from the differential pressure between vertically and horizontally aligned ports of the five-hole probe, respectively. This technique requires careful calibration for each specific aircraft which is carried out using a series of calibration maneuvers (Lenschow and Spyers-Duran, 1989; Kalogiros and Wang, 2002). For a typical TAS of about $100 \, \mathrm{m \, s^{-1}}$ and a sampling rate of a few tens of Hz, commonly used instruments provide spatial resolution of a few meters. A few studies applied fast-response hot-wire or hot-film anemometers to reach better resolution (Sheih et al., 1971; Merceret, 1976a, b; Lenschow et al., 1978) but ensuring the long-term maintenance of those instruments is more difficult.

Although the three components of the wind velocity are measured, many of the subsequent analyses utilize mostly the vertical component to calculate variance and turbulent fluxes (e.g. Nicholls and Leighton, 1986; Tjernström and Rogers, 1996; Faloona et al., 2005; Zheng et al., 2011), which are of primary interest for the structure of the ABL as well as for heat and moisture transport (Stull, 1988). Others estimate turbulence kinetic energy dissipation rate $\epsilon$, which is considered as a practical measure of turbulence strength and as an important parameter for cloud microphysics (Grabowski and Wang, 2013) and turbulence parameterization in mesoscale or global models (Mauritsen et al., 2007). Because the dissipative scales (of the order of millimeters) are hardly resolved in aircraft measurements, the microscopic definition of $\epsilon$ (e.g. Pope, 2000, ch. 5) cannot be directly applied. Instead, the universal scaling of the turbulent velocity statistics in the resolved inertial subrange (Eqs. (1) or (3)) is often exploited in practice to derive $\epsilon$ from moderate resolution airborne measurements (e.g. Lambert and Durand, 1999; Siebert et al., 2006b; Jen-La Plante et al., 2016; Wacławczyk et al., 2020). In such an approach, the assumptions of the theory, including local isotropy and homogeneity, are taken for granted even though in the atmosphere there are directions naturally distinguished in larger scales due to buoyancy and wind shear (e.g. Lenschow, 1974; Darbieu et al., 2015).

A few studies have attempted to consider the limitations of the theory, for example by comparing the estimates of $\epsilon$ derived from the three velocity components independently (e.g. Jen-La Plante et al., 2016). Lothon and Lenschow (2005a, 2007) reported transverse-to-longitudinal ratios of power spectra close to 1 instead of the theoretical 4/3 in DYCOMS-II experiment (Stevens et al., 2003) made with the NSF/NCAR C130 research aircraft (Earth Observing Laboratory) in marine stratocumulus. Lothon and Lenschow (2005b) extended this analysis for several other field experiments made with the same aircraft - GOTEX (Romero and Melville, 2010), IDEAS (Stith and Rogers, 2004), RICO (Rauber et al., 2007b) and EPIC (Raymond et al., 2004) - which covered marine and continental boundary layers, with stratocumulus, cumulus or clear sky conditions. They found the ratios equal to about 0.8 on average but suggested the results might be influenced by the upstream flow distortion. It appears forward of the aircraft due to the air being deflected by the wings and the fuselage when approaching them. After applying a correction for upstream flow distortion due to the wings, $P_w/P_u$ became close to 4/3 on average. However, this correction does not impact $P_v/P_u$ which then remained approximately 0.8. Pedersen et al. (2018) considered the scale-by-scale ratio of horizontal-to-vertical velocity spectra below stratocumulus top for DYCOMS-II and POST (Carman et al., 2012; Malinowski et al., 2013; Gerber et al., 2013) experiments. They found strong scale dependence, with average close to 1 at $\lambda < z_i$ and the values ranging from about 1 to 10 at higher $\lambda$ (see Fig. 2 therein). Nevertheless, there are still rather few works investigating the

proportion between transverse and longitudinal velocity statistics in airborne measurements in the ABL; likely because most estimates of the dissipation rate have been obtained from one wind velocity component only.

Likewise, the scaling exponents in the inertial subrange have not been extensively investigated experimentally in the ABL. Lothon and Lenschow (2005a, 2007, 2005b) reported an average $P_w$ exponent of about -2 instead of the theoretical -5/3 in the five field experiments mentioned above. However, as a result of their upstream flow distortion correction, it became approximately -1.5. The exponents for $P_v$ and $P_u$ averaged about -1.8 and -1.5, respectively. Darbieu et al. (2015) studied the evolution of $P_w$ in turbulence decay during afternoon transition. They observed the slopes of the spectra steeper than theoretical in the

fully convective phase which they potentially related to the role of coherent convective structures and associated anisotropy. On the other hand, they found the slopes gradually flatten during afternoon transition to become considerably flatter than the theoretical around sunset. Nowak et al. (2021) found exponents for both structure functions and power spectra relatively close to the theory in coupled stratocumulus-topped ABL but significantly smaller in absolute values and highly variable with altitude in the decoupled case.

## 1.4    Overview of the current study

The inspiration for this study originates from the surprising results we encountered while analyzing the dissipation rates derived independently from the three wind velocity components measured by an aircraft in a trade-wind ABL. This motivated us to generalize our analysis by focusing on the transverse-to-longitudinal ratio and on the scaling of second order velocity statistics, and by considering other aircraft participating in other field campaigns. Therefore, here we compare the observed ratio of

transverse and longitudinal statistics (structure functions and power spectra) in the inertial subrange with the theoretical value of 4/3. Secondly, we compare the observed scaling of structure functions and power spectra with the theoretical exponents of 2/3 and -5/3, respectively. For this purpose, we use open datasets for four field experiments involving three different aircraft.

The paper is structured as follows. Sec. 2 introduces the measurements of turbulence collected during four field experiments together with the available datasets and explains the selection of data for our study. Sec. 3 describes the methods used to

derive the relevant parameters. Sec. 4 presents the results on the transverse-to-longitudinal ratio and inertial subrange scaling, and compares them with the theoretical predictions. Sec. 5 discusses the possible reasons and consequences of the observed departure from theoretical values. Finally, our findings are summarized in the last section.

## 2    Observations

### 2.1    Field experiments

The measurements considered in this study were performed during four field experiments:

- EUREC4A (Elucidating the role of cloud–circulation coupling in climate) in Jan - Feb 2020 in trade-wind cumulus regime in northwestern Atlantic (Stevens et al., 2021),

- RICO (Rain in Cumulus Over Ocean) in Nov 2004 - Jan 2005 in trade-wind cumulus regime in northwestern Atlantic (Rauber et al., 2007b),

- VOCALS-REx (Variability of the American Monsoon Systems Ocean-Cloud-Atmosphere-Land Study Regional Experiment) in Oct-Nov 2008 in subtropical stratocumulus regime in southeastern Pacific (Wood et al., 2011),

- POST (Physics of the Stratocumulus Top) in Jul-Aug 2008 in subtropical stratocumulus regime in northeastern Pacific (Carman et al., 2012; Malinowski et al., 2013; Gerber et al., 2013).

The objectives, strategy and execution of the experiments are described in the references given above. EUREC4A addressed many research questions comprising atmospheric circulation, clouds, rain formation, life cycle of particulate matter, upper-ocean processes and air-sea interaction. The meteorological conditions and the structure of the ABL observed during EUREC4A are analyzed in detail by Albright et al. (2022). RICO investigated the mechanism of rain formation in shallow cumuli and its feedback on the structure and variability of those clouds. VOCALS-REx studied interactions between aerosols, microphysics, precipitation and radiation in marine stratocumulus as well as physical and chemical couplings between the upper ocean and the lower atmosphere in the region of one of the strongest coastal upwelling. POST focused particularly on processes occurring at the interface between the stratocumulus-topped ABL and the free troposphere.

## 2.2 Aircraft

The turbulence measurements in the ABL analyzed here were obtained with three research aircraft:

- SAFIRE (the French facility for airborne research) ATR42 (SAFIRE) during EUREC4A,

- NSF/NCAR (National Science Foundation - National Center for Atmospheric Research) C130 (Earth Observing Laboratory) during RICO and VOCALS-REx,

- NPS CIRPAS (Naval Postgraduate School - Center for Interdisciplinary Remotely-Piloted Aircraft Studies) Twin Otter (TO; NASA Airborne Science Program) during POST.

The three aircraft are equipped with a five-hole radome probe and the three components of turbulent wind velocity are computed similar to the methods described by Lenschow (1986). The aircraft differ in size and cruising speed. The C130, ATR and TO feature wing span of about 40, 25 and 20 m, respectively. The typical TAS of the ATR is $\sim100\,\mathrm{m\,s^{-1}}$ which with the sampling rate $f_s = 25\,\mathrm{Hz}$ provides a spatial resolution $\Delta r = TAS/f_s \sim 4\,\mathrm{m}$. The TAS, sampling rate and resolution for the C130 is the same as for the ATR. The typical TAS of the TO is $\sim55\,\mathrm{m\,s^{-1}}$ which with $f_s = 40\,\mathrm{Hz}$ gives a resolution $\Delta r \sim 1.4\,\mathrm{m}$. For the ATR during EUREC4A, the instrumentation and sampling strategy are described in Bony et al. (2022) while the turbulence measurements are described in Brilouet et al. (2021).

## 2.3 Datasets

The turbulence data for the four experiments were downloaded from public datasets (Lothon and Brilouet, 2020; UCAR/NCAR - Earth Observing Laboratory, 2011a, b; Khelif, 2009, for EUREC4A, RICO, VOCALS-REx and POST, respectively). For EUREC4A, RICO and VOCALS-REx, turbulent wind velocity is given in the longitudinal-lateral-vertical coordinate system. The longitudinal direction is along the velocity of aircraft with respect to air as explained in sec. 1.1. For POST, turbulent wind velocity is given in the eastward-northward-vertical coordinate system. We computed the longitudinal $u$ and lateral $v$ components by rotating eastward and northward components by the aircraft true heading angle.

## 2.4 Flight segments

We analyze only horizontal flight segments in the ABL. The fixed flight pattern during EUREC4A included straight horizontal segments at four levels: close to the cloud base, near the top of the subcloud layer, in the middle of the subcloud layer and near the surface; in the direction either parallel or perpendicular to the mean wind. During RICO, the flights included horizontal circles (∼60 km diameter) above the surface and below the cloud base as well as straight horizontal segments at various heights in the subcloud and cloud layers. In VOCALS-REx, the flights included straight horizontal segments mostly at ∼100 m or inside the cloud. During POST, the flights included straight horizontal segments typically at three levels in the ABL: close to the cloud top, near the cloud base and near the surface.

For EUREC4A, we applied the definition of segments and their classification according to level (*cloud-base, top-subcloud, mid-subcloud, near-surface*) from the same dataset as turbulence records (Lothon and Brilouet, 2020). In the case of VOCALS-REx, we used segment timestamps and levels (*in-cloud, cloud-base, sub-cloud*) from the related dataset devoted to lidar measurements (Leon et al., 2011).

For RICO and POST, no a priori segment information is available which is why we developed our own segmentation algorithm based on the conditions of small derivatives of altitude and true heading with respect to distance (see Appendix A). We also crudely classified the detected segments according to characteristic levels. The classification is only approximate as the detailed characterization of ABL stratification in each of the flights is beyond the scope of this study. In the case of RICO, we marked the segments below 990 hPa as *near-surface*, others below 950 hPa as *sub-cloud*, others below 900 hPa as *cloud-base*, others below 800 hPa as *cloud-layer*, following Fig. 5 of Rauber et al. (2007b). For POST, we exploited the information on average cloud base height and average cloud top height for each flight from Table 1 of Carman et al. (2012) together with the measurements of liquid water content (LWC) obtained with the particle volume meter (Gerber et al., 1994) available in a separate dataset (Gerber, 2009). The table given in Carman et al. (2012) misses one flight (RF09); hence, for this flight we inferred cloud top and cloud base heights from Table 1 of Gerber et al. (2013). We defined the cloud middle as the height halfway between the base and the top. Based on the LWC, we estimated cloud fraction in each segment as the fraction of data points where LWC>0.02 g m$^{-3}$. The segments below 60 m were marked as *near-surface*. The segments above 60 m and below the cloud middle were considered as *sub-cloud* if the cloud fraction was smaller than 0.5 or *cloud-base* if the cloud fraction

was at least 0.5. The segments above the cloud middle for which the cloud fraction was at least 0.5 were classified as *cloud-top*. The others which did not meet the above criteria were not included in the analysis.

Brilouet et al. (2021) report several technical difficulties encountered during EUREC4A, e.g. concerning one of the radome transducers in flights RF02 to RF08 and the failure of inertial navigation in RF20, and conclude that flights RF09 to RF19 had much better-quality data. For this reason, we considered those 11 flights only. From other experiments, we used all flights available in the datasets.

The segment number, average altitude and length for each experiment and level are summarized in Table 1. In EUREC4A and POST, most of the segments were flown either approximately parallel or perpendicular to the mean wind direction. Hence, we distinguish them in the following figures by filled and open symbols, respectively.

## 3 Analysis

The bulk lateral-to-longitudinal $D_v/D_u$ and vertical-to-longitudinal $D_w/D_u$ ratios of structure functions, and the analogous ratios of power spectral densities $P_v/P_u$, $P_w/P_u$ for each segment were calculated with the methods similar to those used in sec. 4.3 of Nowak et al. (2021) to estimate dissipation rates. Structure functions $D_i$ computed for each velocity component $i = u, v, w$ from linearly detrended records were averaged in 5 logarithmically equidistant bins covering the selected fitting range (defined further). The ratios were obtained by dividing parameters $B_i$ resulting from the least-squares fit of the relationship (c.f. Eq. (1))

$$D_i(r) = B_i r^{2/3}. \tag{5}$$

Power spectral densities $P_i$ were computed from linearly detrended velocity records using the Welch algorithm (Welch, 1967) with window length of $1\,\mathrm{km}$ and window overlap of $0.5\,\mathrm{km}$. Similarly to $D_i$, they were averaged in 5 logarithmically equidistant bins covering the fitting range and the ratios were obtained by dividing parameters $C_i$ resulting from the least-squares fit of the relationship (c.f. Eq. (3))

$$P_i(f) = C_i f^{-5/3} \tag{6}$$

where $f$ is frequency. In addition, we evaluated the scaling exponents of structure functions $s_i$ and power spectra $p_i$ with separate least-squares fits of the formulas

$$D_i(r) = B_i^* r_i^s, \quad P_i(f) = C_i^* f^{-p_i} \tag{7}$$

performed on the same averaged points as for the fits of Eq. (5) and (6). The parameters $B_i^*$, $C_i^*$ are not used in further analysis. The estimation of the uncertainties of the computed transverse-to-longitudinal ratios and scaling exponents is discussed in Appendix B.

The choice of the fitting ranges was guided by the spatial resolution of measurements (see sec. 2.2), integral length scale for the vertical velocity (given in Table 1) and the manual inspection of the observed extension of power-law scaling, in particular

**Table 1.** Statistics of the segments considered in the analysis: number of segments at each level, average altitude, length and integral length scale for vertical wind velocity (defined in sec. 3). Standard deviations are given in parentheses. For EUREC4A and POST, the number of segments is written as the sum of the numbers of segments flown approximately parallel ($\parallel$) and perpendicular ($\perp$) to the mean wind direction.

| Level | Number | Altitude [m] | Integral scale [m] | Length [km] |
|---|---|---|---|---|
| ATR-EUREC4A | | | | |
| cloud-base | $116\perp$ | 806 (83) | 267 (136) | 54 (5) |
| top-subcloud | $11\parallel + 9\perp$ | 592 (45) | 246 (109) | 62 (10) |
| mid-subcloud | $10\parallel + 9\perp$ | 291 (26) | 195 (77) | 56 (9) |
| near-surface | $5\parallel + 5\perp$ | 64 (3) | 58 (30) | 41 (6) |
| C130-RICO | | | | |
| cloud-layer | 53 | 1547 (296) | 258 (224) | 50 (17) |
| cloud-base | 51 | 804 (114) | 164 (144) | 48 (16) |
| sub-cloud | 49 | 399 (70) | 152 (97) | 154 (72) |
| near-surface | 55 | 97 (28) | 81 (23) | 136 (72) |
| C130-VOCALS | | | | |
| in-cloud | 88 | 1156 (265) | 100 (29) | 80 (44) |
| cloud-base | 6 | 570 (216) | 214 (135) | 176 (52) |
| sub-cloud | 84 | 148 (14) | 95 (17) | 76 (44) |
| TO-POST | | | | |
| cloud-top | $9\parallel + 32\perp$ | 443 (140) | 120 (116) | 25 (5) |
| cloud-base | $11\parallel + 11\perp$ | 247 (123) | 101 (109) | 32 (16) |
| sub-cloud | $4\parallel + 38\perp$ | 178 (110) | 55 (43) | 25 (7) |
| near-surface | $4\parallel + 45\perp$ | 32 (6) | 13 (4) | 24 (6) |

for the segments at the lowest levels. The integral length scale $L$ was estimated as the distance at which the autocorrelation function of vertical velocity declines by a factor of $e$ (c.f. Nowak et al., 2021, sec. 4.5). In the case of EUREC4A, RICO and VOCALS-REx, we applied the fitting ranges $[2\Delta r, L]$ for $D_i$ and $[4\Delta r, 2L]$ for $P_i$. The lower ends correspond to twice the smallest $r$ and twice the Nyquist frequency, respectively. For POST, we applied the ranges of $[3\Delta r, L]$ and $[6\Delta r, 2L]$ in order to avoid the influence of a spurious peak at $\sim 5.5\,\mathrm{m}$ corresponding to the frequency of $\sim 10\,\mathrm{Hz}$ which is symptomatic for most of the segments. Figures 1 and 2 shows structure functions and power spectra, respectively, together with universal scaling

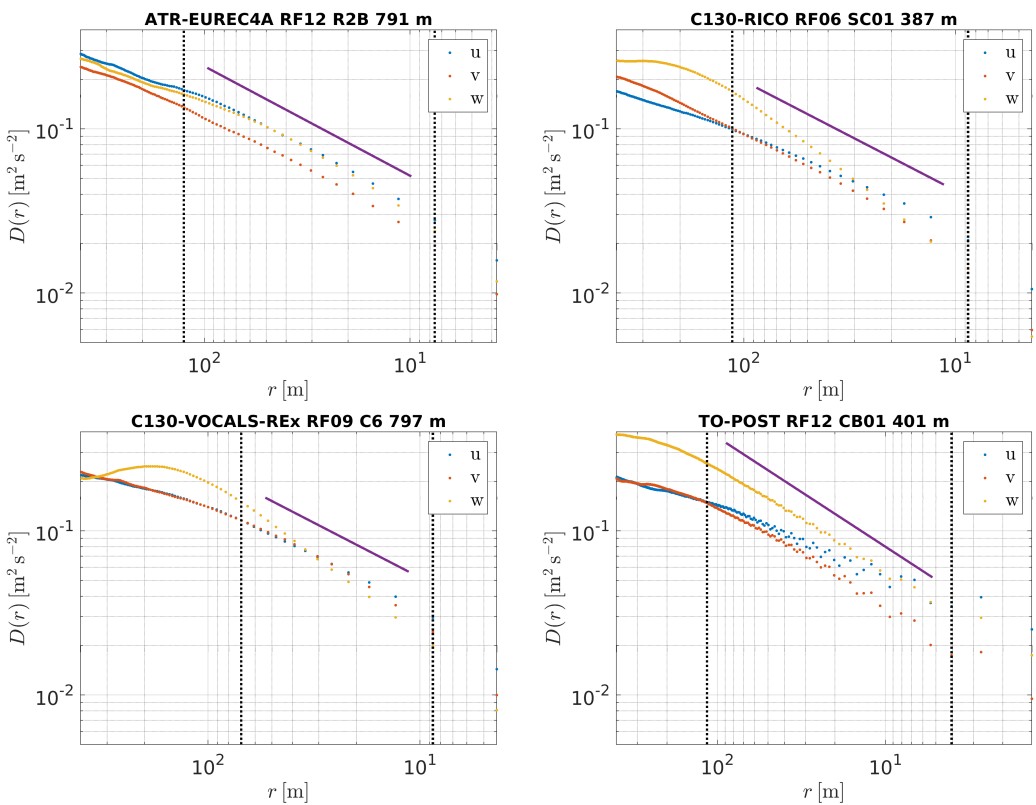

**Figure 1.** Structure functions for single segments from each experiment. The aircraft, experiment, flight, segment name and segment altitude are given in panel titles. The purple solid line denotes the universal 2/3 scaling. The vertical black dotted lines mark the extent of the selected fitting range. Note the orientation of the horizontal axis is from large to small scales.

reference and corresponding fitting range for single segments from each of the four experiments. The sensitivity of the results to the choice of the fitting range is discussed in Appendix C.

The different fitting ranges for $D_i$ and $P_i$ are used here following the remarks given by Chamecki and Dias (2004) and Wacławczyk et al. (2020). The former found a shorter extension of the inertial subrange in the structure functions in comparison with the power spectra (which manifests in diverging ratios $D_T/D_L$ and $P_T/P_L$). The latter derived $\epsilon$ with inertial scaling methods and found the best agreement with reference $\epsilon$ for the structure function fitting range moved towards smaller scales in comparison with the fitting range for power spectra. We observed that the power-law in power spectra typically extends to scales larger than our estimation of the integral length scale.

The scale-by-scale ratios of structure functions and power spectral densities were calculated similarly to sec. 4.b of Siebert et al. (2006b) and sec. 4.4 of Nowak et al. (2021). The relevant statistics were first averaged in logarithmically equidistant bins covering the entire available range of scales (not only the fitting range as before), and the ratios were then computed point-by-point. This procedure is illustrated in Fig. 3 for an example segment. In order to obtain composite scale-by-scale ratios at

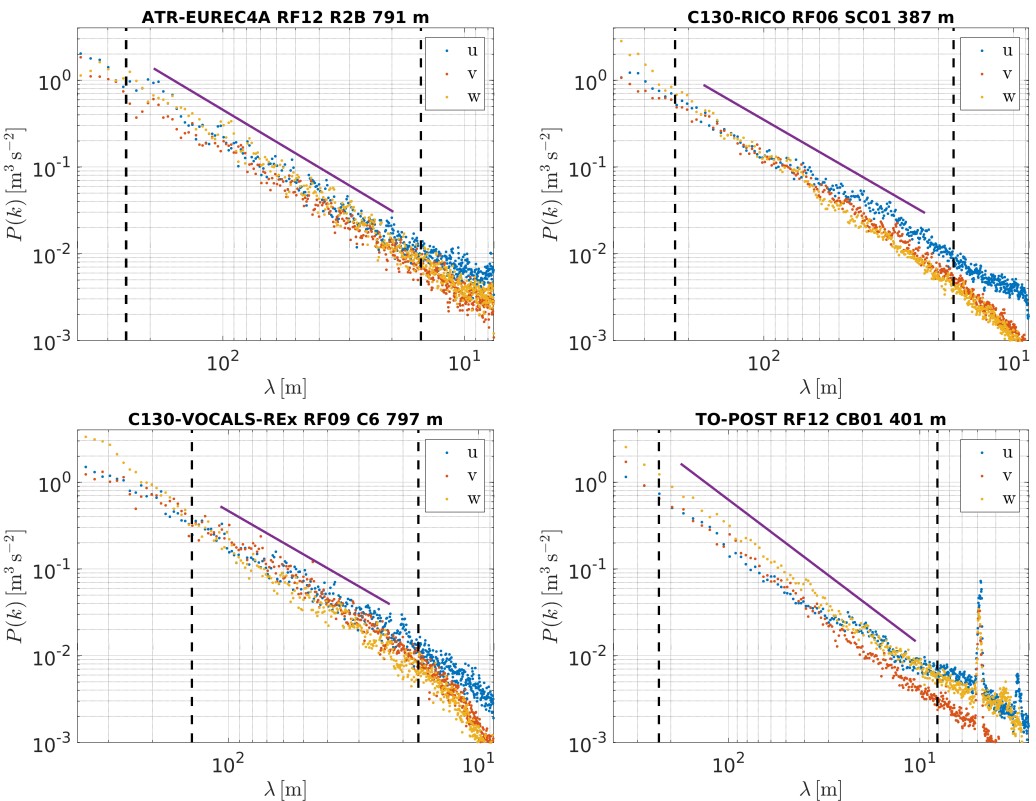

**Figure 2.** Power spectral densities for single segments from each experiment. The aircraft, experiment, flight, segment name and segment altitude are given in panel titles. The purple solid line denotes the universal -5/3 scaling. The vertical black dashed lines mark the extent of the selected fitting range.

the characteristic levels (c.f. Table 1), the single-segment results, as those in the right panel of Fig. 3, were first interpolated to fixed $r/L$ or $\lambda/L$ grid, and the interpolated values were then averaged among the segments at each normalized scale.

## 4  Results

The bulk lateral-to-longitudinal ratios $D_v/D_u$, $P_v/P_u$ are presented in Fig. 4. In general, most of the points cluster in the vicinity of the value of 3/4 for both ratios, in particular in the case of EUREC4A. This stands in striking contrast with 4/3 predicted for homogeneous isotropic turbulence. The largest variability is observed for POST, the smallest for EUREC4A. The former is likely connected with the segment lengths shorter than for other experiments which increases random error (c.f. Lenschow et al., 1994, Eq. (36)), relatively shallow ABL depth and strong wind shear at cloud top (Carman et al., 2012; Malinowski et al., 2013; Jen-La Plante et al., 2016). Nevertheless, $D_v/D_u$ and $P_v/P_u$ approximately agree with each other in all the experiments. There are only minor differences between the levels within the experiments, see the average values reported

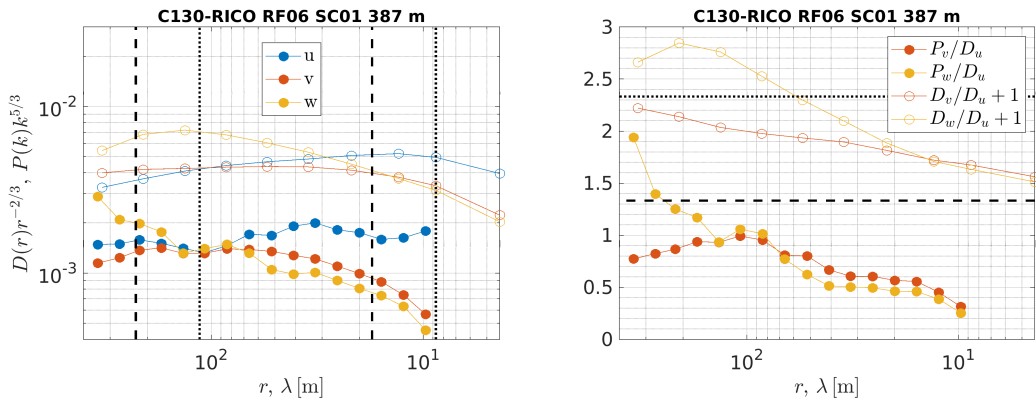

**Figure 3.** Calculations of scale-by-scale transverse-to-longitudinal ratios of structure functions (open circles) and power spectra (filled circles) for a single segment. Left panel shows averaged and compensated statistics together with corresponding fitting ranges from Fig. 1 (black dotted lines) and 2 (black dashed lines). Right panel shows their transverse-to-longitudinal ratios. Those for structure functions are shifted by 1 for clarity. The black horizontal lines mark the isotropic value.

in Table 2. The level averages range from 0.67 to 0.97. The experiment averages range from 0.72 to 0.94 which is 30-46 % smaller than the theoretical value. The experiment-averaged lateral-to-longitudinal ratio is the largest for VOCALS-REx and the smallest for POST. The average $P_v/P_u$ values are roughly in agreement with Lothon and Lenschow (2005b) for all the experiments and levels.

The bulk vertical-to-longitudinal ratios $D_w/D_u$, $P_w/P_u$ are shown in Fig. 5. Almost all of the points are far from the predicted 4/3 value. The largest variability is observed for POST and RICO, the smallest for EUREC4A. In contrast to the lateral-to-longitudinal ratios, the differences between the aircraft are more significant. Apart from distinct variability, there is little difference between RICO and VOCALS-REx which both involved the C130. For EUREC4A and POST, $D_w/D_u$ approximately agrees with $P_w/P_u$. For RICO and VOCALS-REx, $D_w/D_u$ is systematically higher than $P_w/P_u$. There are

also some variations between the levels (see the averages given in Table 2), possibly due to the impact of buoyancy or mean wind shear (c.f. Darbieu et al., 2015; Pedersen et al., 2018; Akinlabi et al., 2019). For example, on average the mid-subcloud level exhibits higher ratios than other levels for EUREC4A while the near-surface level is characterized by lower ratios than cloud levels for POST and RICO. The level averages range from 0.80 to 1.11 for $D_w/D_u$ and from 0.64 to 1.10 for $P_w/P_u$ which is 16-40 % and 17-52 % smaller than the theoretical value.

Fig. 6 presents the exponents $s$ and $p$. The points are dispersed in the neighborhood of the predicted $s = 2/3$ and $p = 5/3$. There are considerable differences between velocity components. The clusters of points representing the longitudinal component are almost separated from those for the transverse components in the case of RICO, VOCALS-REx and POST. The differences related to the aircraft are also visible. The variations among the levels within the experiments are rather minor. Hence, we report the average values for entire experiments in Table 3. The experiment-averaged structure functions exponents

can be from 0.44 for $s_u$ to 0.98 for $s_w$, i.e. 34 % lower and 47 % higher than the predicted 2/3. The experiment-averaged power

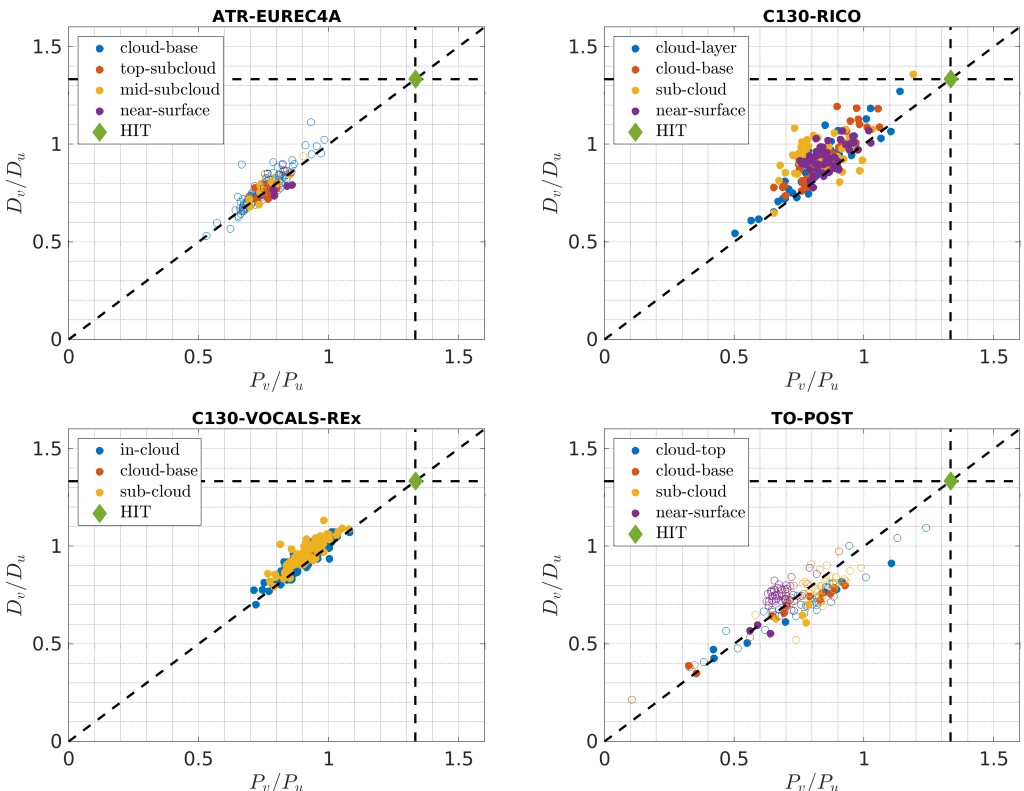

**Figure 4.** The bulk lateral-to-longitudinal ratios of structure functions with respect to the analogous ratios of power spectra. Each circle denotes one segment. For EUREC4A and POST, filled and open symbols correspond to the segments flown parallel and perpendicular to the mean wind direction, respectively. Colors denote characteristic levels of the boundary layer (see sec. 2.4 and Table 1). Horizontal and vertical black dashed lines mark the value 4/3. Diagonal black dashed line denotes 1:1 proportion. The green diamond shows the theoretical prediction for homogeneous isotropic turbulence (HIT).

spectra exponents take values from 1.26 for $p_u$ to 2.03 for $p_w$ which is 24 % lower and 22 % higher than 5/3. Particularly close to the theoretical predictions are the average exponents for EUREC4A, $s_u$ for RICO as well as $s_v$ and $p_v$ for POST. For RICO and VOCALS, average $p_w$ is close to 2 in agreement with the results of Lothon and Lenschow (2005a, b, 2007) before applying their upstream flow distortion correction.

The composite scale-by-scale lateral-to-longitudinal ratios are presented in Fig. 7 for the range of scales from about $0.01L$ to $3L$. The ratios are significantly smaller than 4/3 throughout those scales, except only for the largest $3L$ in the case of RICO, VOCALS-REx and POST. This is true for the composites as well as for majority of the individual segments, which is illustrated by the shaded range defined by standard deviation. For clarity, the shading is shown for only one level in each experiment but the standard deviations for other levels are of the same order. Importantly, all the curves exhibit the same overall trend, decreasing

and increasingly departing from 4/3 with decreasing scale. This trend corresponds well to the scalings of $D_v$ and $P_v$ which are

**Table 2.** Average values of the ratios of structure functions and power spectra. Standard deviations are given in parentheses.

| Level | $D_v/D_u$ | $P_v/P_u$ | $D_w/D_u$ | $P_w/P_u$ |
|---|---|---|---|---|
| ATR-EUREC4A | | | | |
| cloud-base | 0.77 (0.09) | 0.75 (0.08) | 0.87 (0.11) | 0.87 (0.12) |
| top-subcloud | 0.77 (0.05) | 0.75 (0.04) | 0.88 (0.10) | 0.87 (0.10) |
| mid-subcloud | 0.78 (0.06) | 0.77 (0.05) | 0.98 (0.08) | 0.95 (0.05) |
| near-surface | 0.80 (0.05) | 0.81 (0.03) | 0.84 (0.05) | 0.88 (0.05) |
| all | 0.78 (0.08) | 0.76 (0.07) | 0.88 (0.10) | 0.88 (0.11) |
| C130-RICO | | | | |
| cloud-layer | 0.89 (0.14) | 0.82 (0.12) | 1.11 (0.28) | 0.84 (0.19) |
| cloud-base | 0.93 (0.12) | 0.85 (0.09) | 1.11 (0.15) | 0.88 (0.11) |
| sub-cloud | 0.95 (0.10) | 0.84 (0.10) | 0.97 (0.15) | 0.64 (0.12) |
| near-surface | 0.93 (0.06) | 0.85 (0.06) | 0.91 (0.12) | 0.68 (0.10) |
| all | 0.92 (0.11) | 0.84 (0.10) | 1.02 (0.21) | 0.76 (0.17) |
| C130-VOCALS | | | | |
| in-cloud | 0.91 (0.07) | 0.89 (0.07) | 0.95 (0.13) | 0.77 (0.08) |
| cloud-base | 0.97 (0.03) | 0.94 (0.04) | 0.91 (0.13) | 0.81 (0.11) |
| sub-cloud | 0.96 (0.07) | 0.91 (0.06) | 0.92 (0.12) | 0.75 (0.06) |
| all | 0.94 (0.07) | 0.90 (0.07) | 0.94 (0.13) | 0.76 (0.07) |
| TO-POST | | | | |
| cloud-top | 0.70 (0.15) | 0.75 (0.19) | 1.04 (0.24) | 1.10 (0.31) |
| cloud-base | 0.67 (0.18) | 0.69 (0.22) | 0.90 (0.20) | 0.95 (0.26) |
| sub-cloud | 0.77 (0.09) | 0.81 (0.08) | 0.94 (0.13) | 1.08 (0.14) |
| near-surface | 0.75 (0.08) | 0.68 (0.08) | 0.80 (0.09) | 0.85 (0.09) |
| all | 0.73 (0.12) | 0.74 (0.15) | 0.92 (0.19) | 0.99 (0.23) |

steeper than for $D_u$ and $P_u$ (c.f. Fig. 6 and Table 3). It is apparently the weakest in the case of EUREC4A where the difference in scaling exponents between $v$ and $u$ is the smallest. Moreover, the observed scale dependence is comparable for different levels of the ABL, except for the near-surface, which might be influenced by wind shear and where the integral length scales are substantially smaller than at other levels (see Table 1).

The composite scale-by-scale vertical-to-longitudinal ratios in Fig. 8 show the features similar to the lateral-to-longitudinal ratios. They are mostly smaller than 4/3, except for largest scales. With decreasing scale, they exhibit the overall decrease and

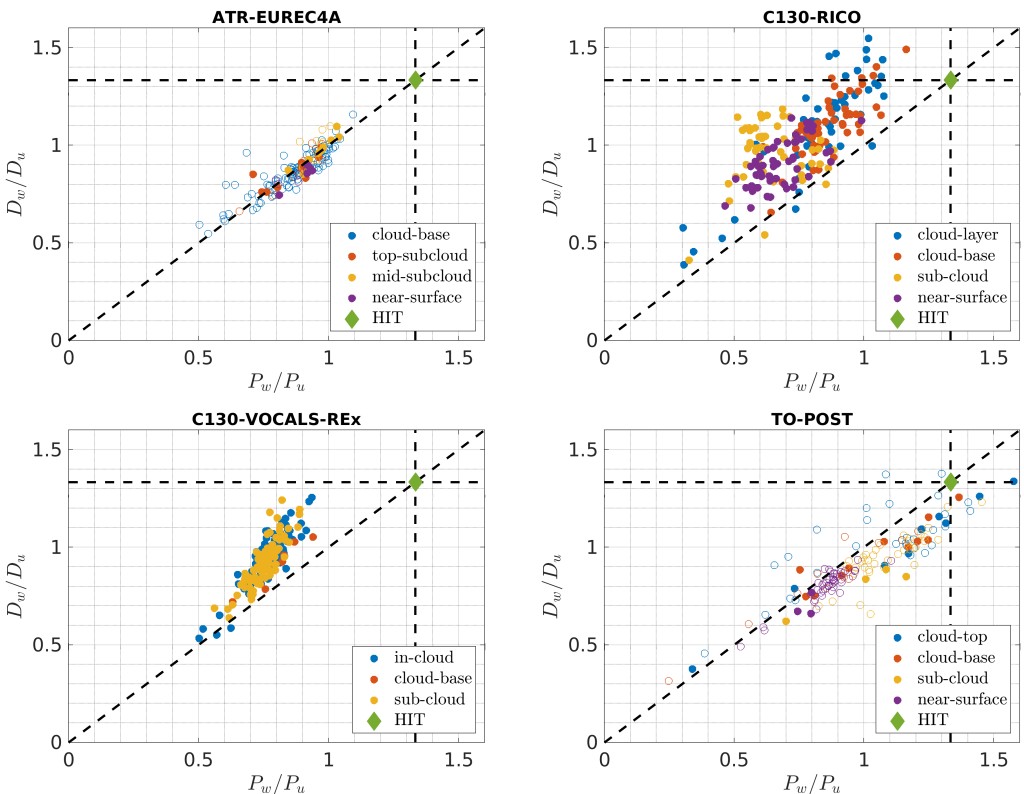

**Figure 5.** As in Fig. 4 but for the vertical-to-longitudinal ratios. One point for RICO (in cloud layer) and one point for POST (at cloud top) lie outside the range presented here.

**Table 3.** Average values of the scaling exponents of structure functions and power spectra. Standard deviations are given in parentheses.

| Aircraft/Campaign | $s_u$ | $s_v$ | $s_w$ | $p_u$ | $p_v$ | $p_w$ |
|---|---|---|---|---|---|---|
| ATR-EUREC4A | 0.67 (0.05) | 0.72 (0.07) | 0.69 (0.07) | 1.66 (0.11) | 1.68 (0.11) | 1.71 (0.08) |
| C130-RICO | 0.64 (0.06) | 0.80 (0.07) | 0.98 (0.11) | 1.55 (0.10) | 1.78 (0.08) | 2.03 (0.13) |
| C130-VOCALS | 0.62 (0.04) | 0.75 (0.04) | 0.93 (0.08) | 1.51 (0.06) | 1.80 (0.06) | 1.93 (0.09) |
| TO-POST | 0.44 (0.05) | 0.71 (0.12) | 0.59 (0.09) | 1.26 (0.14) | 1.66 (0.11) | 1.55 (0.17) |

increasing departure from 4/3 in agreement with the derived scaling exponents. The curves at different levels are of similar shape but vary in magnitude more than the lateral-to-longitudinal ratios. Interestingly, particularly high values are reached at the largest scales in the case of RICO, which might be associated with cumulus convection containing strong vertical updrafts.

The increasing departure from isotropy with decreasing scale is in striking contrast to the investigations on the onset of local isotropy in the surface layer (Kaimal et al., 1972; Katul et al., 1997; Siebert and Muschinski, 2001; Chamecki and Dias,

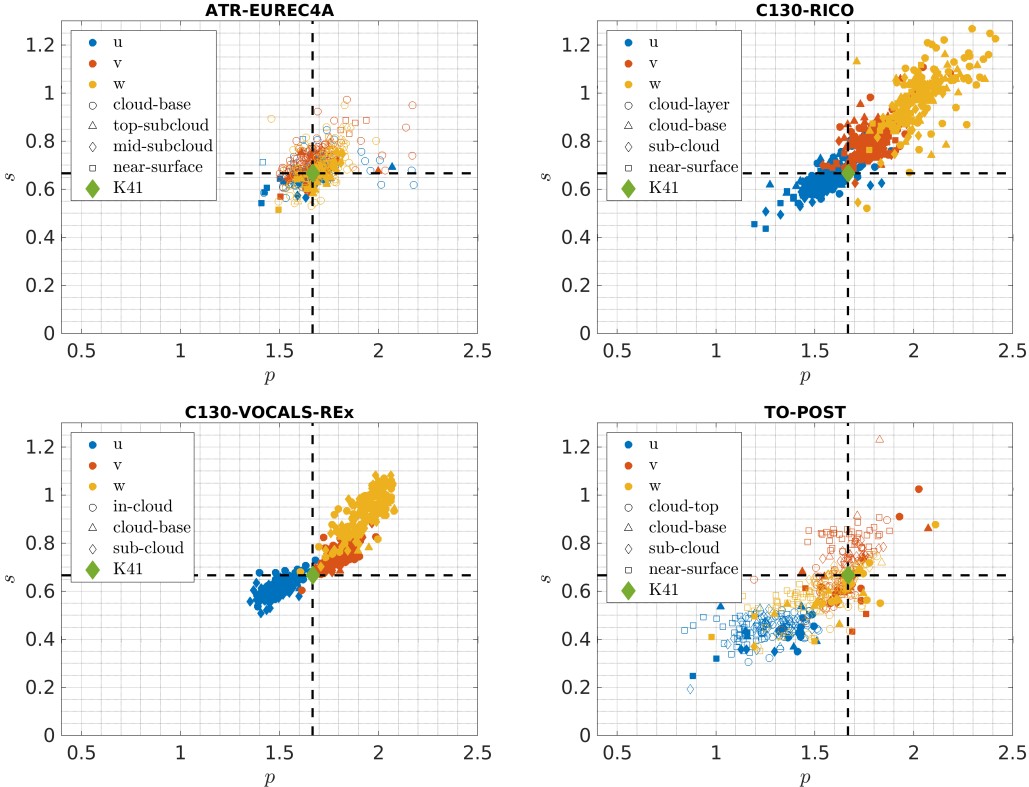

**Figure 6.** The exponents of structure functions $s$ with respect to the exponents of power spectra $p$. Each circle denotes one segment. For EUREC4A and POST, filled and open symbols correspond to the segments flown parallel and perpendicular to the mean wind direction, respectively. Colors denote velocity components while different symbols denote characteristic levels of the boundary layer (see sec. 2.4 and Table 1). Black dashed lines mark the theoretical values of 2/3 and 5/3. The green diamond shows the prediction of the Kolmogorov theory (K41).

2004), who found that the local isotropy is gradually approached with decreasing scale, and with the studies on scale-by-scale anisotropy above the surface layer (Kaimal et al., 1976, 1982; Siebert et al., 2006b; Pedersen et al., 2018; Nowak et al., 2021), who found the transverse-to-longitudinal ratios relatively close to 4/3 at least in some range of scales unaffected by instrumental
deficiencies.

## 5 Discussion

The results of our analysis suggest that the variability in the transverse-to-longitudinal ratios and scaling exponents of velocity statistics can be attributed to how the velocity components are measured on the aircraft. The differences between field experiments and ABL levels seem to be of secondary importance. This motivates an examination of the details of measure-
355 ment technique and instrument properties. In general, airborne measurements suffer from errors which are often challenging

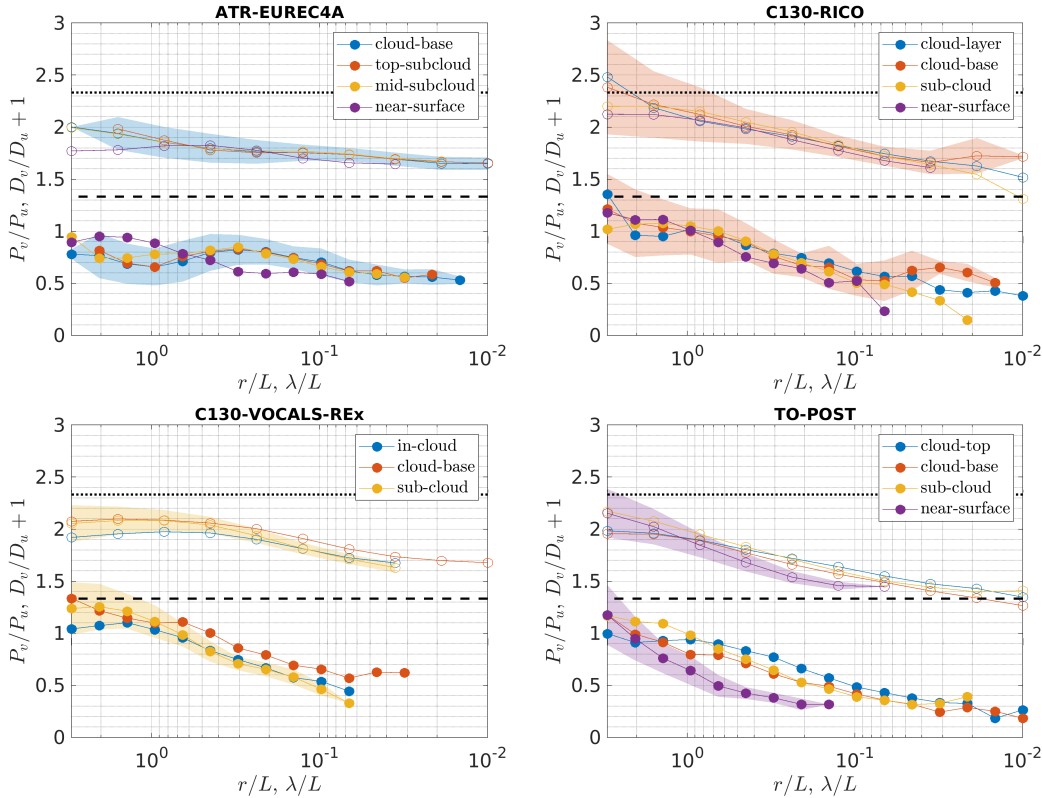

**Figure 7.** The composite scale-by-scale lateral-to-longitudinal ratios for structure functions (open circles) and power spectra (filled circles). Color denote characteristic levels of the boundary layers (see sec. 2.4 and Table 1). The results for structure functions are shifted by 1 for clarity. The horizontal black dotted and dashed lines mark the isotropic value for shifted structure functions and power spectra, respectively. The shading illustrates the range of +/- one standard deviation among the segments averaged to obtain the composite ratios. For clarity, it is drawn only for a selected level for each experiment: cloud-base for EUREC4A, cloud-base for RICO, sub-cloud for VOCALS-REx and near-surface for POST.

to quantify because of flow distortion induced by the airplane (Wendisch and Brenguier, 2013). Rauber et al. (2007a) reported that velocity measurements on the C130 during RICO showed attenuation at high frequencies for $v$ and $w$. The measurements for VOCALS-REx probably suffered from the same issue. This can be spotted in the spectra in Fig. 2 which are representative of most of the segments. In contrast to Rauber et al. (2007a), we observe $w$ to be more affected than $v$. A similar problem

is evident for the TO during POST. In addition, the POST spectra exhibit a pronounced peak at $\sim 5.5\,\mathrm{m}$ corresponding to a frequency of $\sim 10\,\mathrm{Hz}$ which is symptomatic for most of the segments. The peak may have resulted from an internal resonance of the measurement system (Djamal Khelif, personal communication). However, this effect influences the wavelengths outside our fitting range, so it does not explain the results, in particular the departure of the transverse-to-longitudinal ratios from the predicted 4/3.

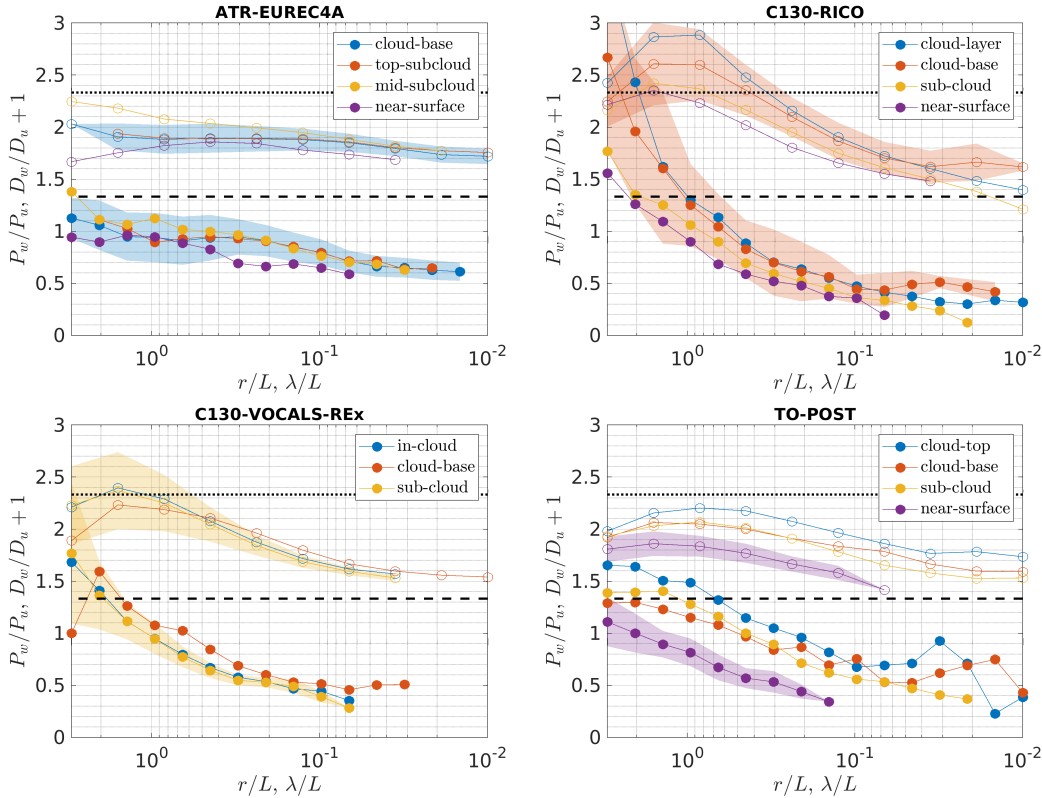

**Figure 8.** As in Fig. 7 but for the vertical-to-longitudinal ratios.

The vertical-to-longitudinal ratios might be affected by the environmental conditions violating the isotropy assumption, mostly related to the impact of buoyancy. Our analysis involves measurements performed in the convective ABLs under shallow trade-wind cumulus and subtropical stratocumulus regimes. The circulation inside both types of ABL is driven by buoyancy: primarily by negative buoyancy induced by longwave radiative cooling at stratocumulus top (Wood, 2012) and positive buoyancy due to surface heat fluxes in the trade-wind subcloud layer (Albright et al., 2022). Both situations lead to

positive buoyancy flux across most of the mixed layer (parcels of negative buoyancy descend from the top while those with positive buoyancy rise from the surface). In general, the influence of buoyancy on turbulence anisotropy depends on the sign of buoyancy flux, as documented e.g. in the direct numerical simulation of stratocumulus by Akinlabi et al. (2019). Inside the cloud interior they found $P_w$ of higher magnitude than predicted assuming local isotropy ($C_T \approx 1$, not 0.65). Such an excess of energy in $w$ was attributed to buoyant forcing which favours vertical motions and pressure redistribution apparently insufficient

to isotropize turbulence. At the very cloud top, $P_w$ was strongly weakened with respect to isotropic prediction due to stable stratification and corresponding negative buoyancy flux consuming kinetic energy. This implies $P_w/P_u > 4/3$ inside the cloud interior and $P_w/P_u < 4/3$ at its very top. Nowak et al. (2021) also observed $P_w/P_u \gtrsim 4/3$ in a stratocumulus-topped ABL

within a limited range of scales. They speculated that those scales might represent typical horizontal sizes of surface layer plumes or cloud top downdrafts (see also sec. 1.3).

In addition to buoyancy, wind shear can also modify anisotropy of turbulence by strengthening motions in a specific direction. For example, Akinlabi et al. (2019) found that large-scale flow instabilities induced by shear enhanced $P_u$, yet only at relatively large scales. Note that a similar idea of interplaying impacts of buoyancy and shear applies also to the surface layer. As mentioned in sec. 1.2, Katul et al. (1995) suggested that under stable conditions buoyancy and shear superimpose in maintaining anisotropy but under unstable conditions they counteract resulting in more isotropic turbulence.

Nevertheless, although buoyancy and wind shear certainly affect the character of turbulence, it is unlikely these factors explain our results. The computed $D_w/D_u$, $P_w/P_u$ are smaller than 4/3 at all levels of the ABL (see Table 2) and almost all considered scales. Even if there was very strong wind shear, it should be concentrated near the surface and the top of the ABL. Also, the substantial deviations of $D_v/D_u$, $P_v/P_u$ from 4/3 are hardly possible to justify either with instrumental factors or boundary conditions as they exist even in the interior (far from the surface and top) of well-mixed ABL. Note that the uncertainties are also smaller than those deviations, see Appendix B.

Consequently, the reason for the disagreement between the observations and the theory remains uncertain. Though, we presume that potential explanation might be the uncertain influence of the flow around an airplane which has finite mass and complex geometry (e.g. upstream flow distortion). This issue deserves attention and further investigation, which would likely help us improve our measurements of turbulence.

In particular, the documented departure of the transverse-to-longitudinal ratio from the predicted isotropic value directly relates to the disparate estimates of dissipation rate obtained separately for three wind velocity components using the universal scaling as in Eqs. (1) and (3). We suggest a way to solve this problem might be to carry out a study of the turbulence energy budget throughout the ABL with an airplane equipped with the radome-based measuring system using a flight and analysis strategy similar to that used by Lenschow (1974). This would be best carried out over a flat homogeneous surface in a situation of strong surface heating and light wind to maximize the ratio of buoyancy production of turbulence to shear production. By flying a series of horizontal flight legs at several levels throughout the ABL, the total production of turbulence within the ABL can be quantified from the integrated buoyancy flux, plus possibly a small contribution from the shear production term near the surface, and compared to the total dissipation integrated throughout the ABL separately using all three wind component measurements to see which gives the best results. We also think that the longitudinal component is most likely to give correct dissipation measurements since it is less affected by flow distortion and has a long history of use on many aircraft in many studies of atmospheric turbulence.

Another approach would be to compare aircraft measurements to measurements at the same height from a tall tower over a horizontally homogeneous surface. An example where this strategy was carried out is given by Kaimal et al. (1982), where turbulence measurements from a 300 m tower (the Boulder Atmospheric Observatory, which no longer exists) were compared with measurements at 150 m and 300 m from a light twin-engine aircraft. They found good agreement among all the wind components in the inertial subrange but in this case the transverse wind components on the aircraft were measured with vanes

at the tip of a 3 m nose boom instead of a typical five-hole probe, which suggests that this may be an issue with the radome technique, and the comparison was carried out over gently rolling terrain.

Moreover, numerical modeling can be beneficial for quantifying the influence of flow distortion on the measurement of turbulent velocity with a five-hole probe located on the aircraft nose. For instance, large eddy simulations of the flow around a popular model of an ultrasonic anemometer helped discern flow distortion errors depending on the azimuth angle and the frequency of velocity variations (Huq et al., 2017). Numerical experiments are particularly important in situations where no laboratory or wind tunnel characterization is possible, such as with a true-size aircraft nose. However, an adequate model needs to be applied in order to account for compressibility which may become important at inflow velocities relevant for typical aircraft.

## 6 Summary

The classical theory of homogeneous isotropic turbulence predicts the ratios of transverse to longitudinal second order velocity structure functions and power spectra are 4/3 in the inertial subrange. In the inertial subrange, those statistics should exhibit power-law scaling with an exponent of +2/3 and -5/3 for the structure functions and power spectra, respectively.

We studied the transverse-to-longitudinal ratios and scaling exponents derived from high-rate pressure in-situ measurements performed by three research aircraft (SAFIRE ATR42, NSF/NCAR C130, CIRPAS Twin Otter), all equipped with a high-rate five-hole radome probe, during four field experiments (EUREC4A, RICO, VOCALS-REx, POST) in two regimes of the marine atmospheric boundary layer (shallow trade-wind convection and subtropical stratocumulus).

The observed lateral-to-longitudinal ratios $D_v/D_u$, $P_v/P_u$ significantly depart from the theoretical value. The experiment-averaged values are from 0.73 to 0.94 which is 30-46 % smaller than predicted. The differences between the levels of the ABL are hardly noticeable. There is a good agreement of $D_v/D_u$ with $P_v/P_u$.

The vertical-to-longitudinal ratios $D_w/D_v$, $P_w/P_u$ exhibit higher variability. They also depart from 4/3. There are significant differences between the aircraft and some noticeable variations between the characteristic levels. Despite different ABL regime, there is little difference between RICO and VOCALS-REx which both involved C130. The level averages are from 0.64 to 1.11 which is 16-52 % smaller than predicted.

On the other hand, the scaling exponents $s$ and $p$ are for the most part distributed around Kolmogorov's 2/3 and 5/3 power law exponents, respectively. The experiment averages differ from the predicted values by -34 to +47 % for structure functions and by -24 to +22 % for power spectra. There are significant differences between aircraft, and between longitudinal and transverse wind velocity components. The variations among the levels are minor. The results for RICO and VOCALS-REx are similar in spite of a different ABL regime.

The composite scale-by-scale transverse-to-longitudinal ratios generally decrease and increasingly depart from 4/3 with decreasing scale, in contrast to previous studies on local isotropy. The curves exhibit similar shapes but can vary in magnitude among the considered levels of the ABL.

In general, our results suggest that the variability in the transverse-to-longitudinal ratios and scaling exponents can be attributed to how the velocity components are measured on the aircraft. The differences between field experiments, representing different ABL regimes, and between ABL levels are of secondary importance. The explanation of the large departures of the transverse-to-longitudinal ratio from 4/3 remains uncertain. This issue warrants further investigation as it is currently a major impediment in using aircraft measurements to study the structure of atmospheric turbulence.

*Code and data availability.* The data used in this study were downloaded from the public datasets (Lothon and Brilouet, 2020; UCAR/NCAR - Earth Observing Laboratory, 2011a, b; Khelif, 2009; Leon et al., 2011; Gerber, 2009). The MATLAB code we developed for the purpose of the presented analysis is available in the repository Nowak et al. (2024).

## Appendix A: Segmentation algorithm

In order to select horizontal segments in RICO and POST flights (see sec. 2), we designed a simple algorithm which exploits the timeseries of altitude $z$, true heading $\psi$ and TAS. The conditions are small derivatives of altitude $dz/dx$ and true heading $d\psi/dx$ with respect to distance $x$ as well as large TAS. The continuous flight legs where all samples meet those conditions constitute segments. From such a set of segments, we take only those exceeding the minimum length (specified below) and with a small overall altitude trend.

The C130 and TO differ in size, cruising speed and other airplane properties. Moreover, RICO flight strategy utilized large circles at constant altitude whereas POST utilized straight segments. Therefore, we separately adjusted the thresholds for those experiments. For C130 during RICO, we required: $4\,\mathrm{km}$ moving average of $dz/dx$ smaller than $10\,\mathrm{m\,km^{-1}}$, $20\,\mathrm{km}$ moving average of $d\psi/dx$ smaller than $3\,^\circ\mathrm{km^{-1}}$, and segment length larger than $30\,\mathrm{km}$. For TO during POST, we required: $2\,\mathrm{km}$ moving average of $dz/dx$ smaller than $12\,\mathrm{m\,km^{-1}}$, $2\,\mathrm{km}$ moving average of $d\psi/dx$ smaller than $5\,^\circ\mathrm{km^{-1}}$, and segment length larger than $20\,\mathrm{km}$. In both cases, the minimum acceptable TAS was $0.9$ of its flight median and the maximum acceptable altitude trend within the segment was $2\,\mathrm{m\,km^{-1}}$. An illustration of the segmentation algorithm applied to one of the RICO flights is given in Fig. A1.

## Appendix B: Uncertainties

We did not consider the errors for the individual instruments onboard research aircraft because the contributions to the final measurement error related to the characteristics of the flow around the fuselage and the environmental conditions are often significant but hardly possible to quantify accurately. Instead, we evaluated the standard errors of the least-squares fits of the formulas in Eqs. (5), (6) and (7). Those errors are indirectly affected by the integral length scale estimates which control the width of the fitting range.

The uncertainties in the presented results, i.e. the transverse-to-longitudinal ratios and scaling exponents, are obtained from appropriately propagated errors originating from least-squares fits. We show their ranges in the form of box-and-whisker plots

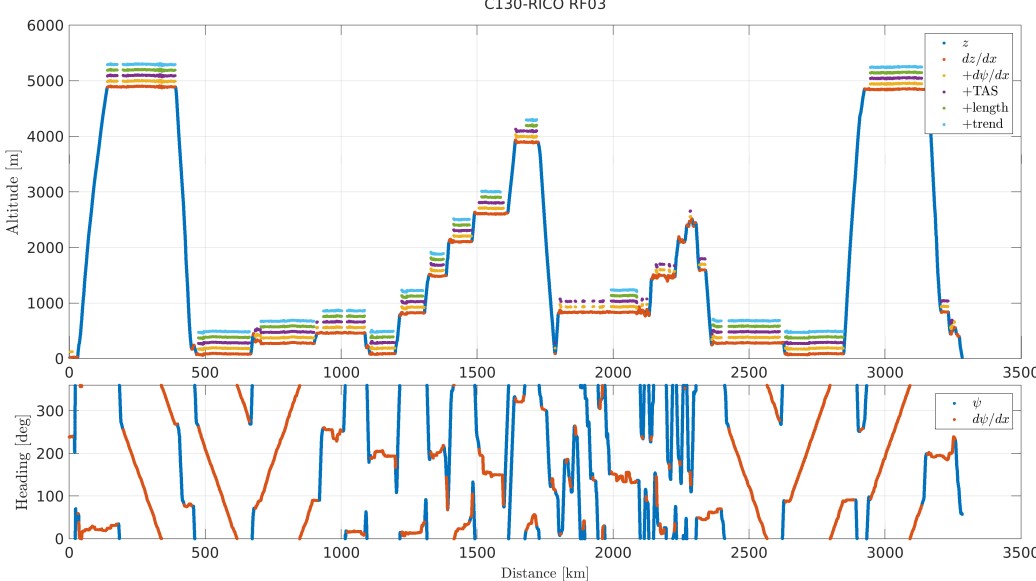

**Figure A1.** Segmentation algorithm applied to C130 RICO flight RF03. Upper panel: altitude (blue), sample points meeting the criteria of: small $dz/dx$ (red), also small $d\psi/dx$ (yellow), also large TAS (purple), also large segment length (green), also small overall altitude trend (cyan). Lower panel: true heading (blue), samples meeting the criterion of small $d\psi/dx$ (red).

in Fig. B1. For the transverse-to-longitudinal ratios, the median values are below 0.2. In general, the lowest uncertainties are
observed for EUREC4A while the highest for RICO and POST. The median uncertainties of $s$ and $p$ are below 0.05 and 0.1, respectively. Here, there is no clear tendency with respect to the experiment.

### Appendix C:  Sensitivity to fitting range

We examined the sensitivity of the results with respect to the choice of the fitting range by repeating the computations of the transverse-to-longitudinal ratios and the scaling exponents for six different values for the upper end of this range: from $0.6L$
to $1.4L$ separation distance in the case of structure functions and from $1.2L$ and $2.8L$ wavelength in the case of power spectra. The upper end for power spectra was twice as large as for structure functions in each such test. The other parameters, including the lower end of the fitting range, were kept the same as given in Sec. 3.

The results were not observed to change significantly with the fitting range. The plots as in Figs. 4, 5 and 6 are to a large extent similar regardless of the considered fitting range (not shown). In Fig. C1 we present the experiment-averaged results
for each test. The variations related to the changes in the fitting range are typically smaller than between the experiments and negligible in comparison to the variability among individual segments visible in Figs. 4-6.

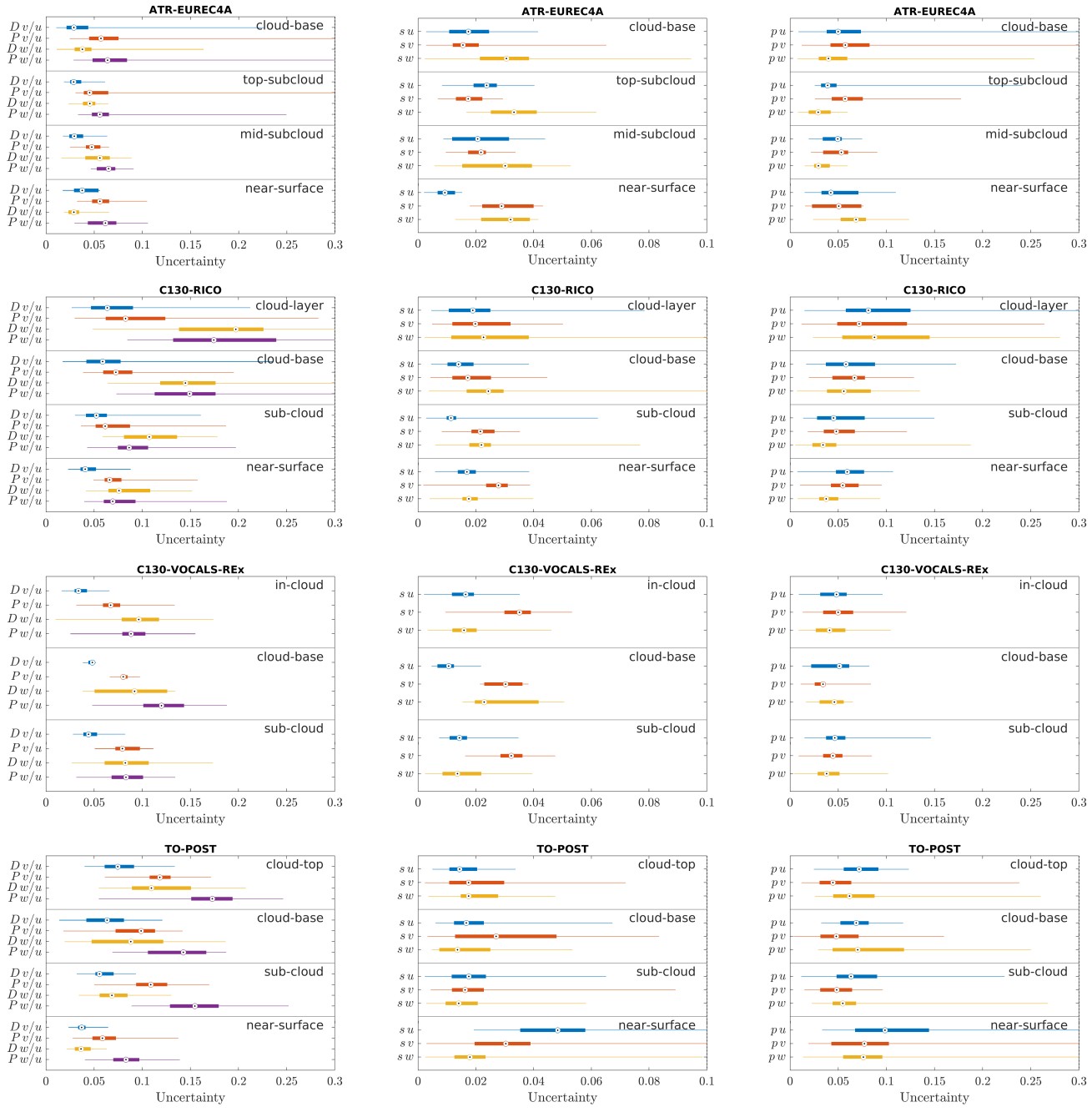

**Figure B1.** Uncertainties of the transverse-to-longitudinal ratios and scaling exponents for structure functions and power spectra in the form of box-and-whisker plots illustrating the range of values among segments belonging to each level in each experiment. The dot inside the box denotes the median value, box spans the interquartile range and whiskers span the entire range.

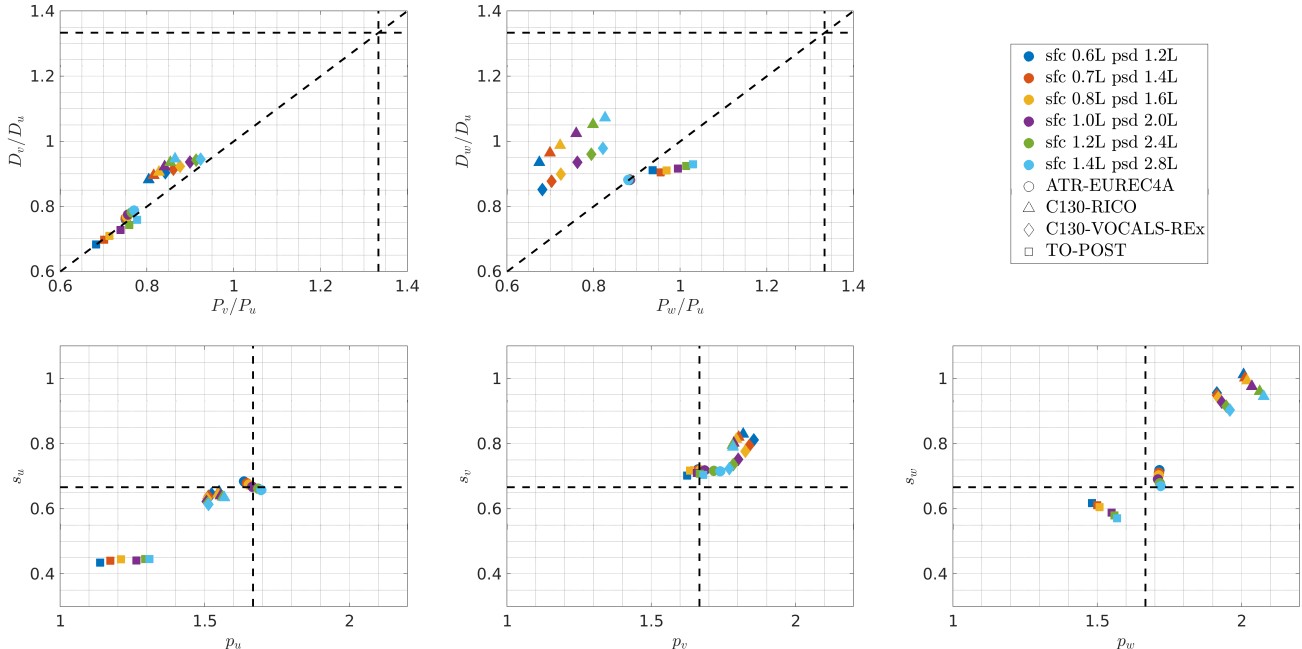

**Figure C1.** Experiment-averaged results on transverse-to-longitudinal ratios and scaling exponents obtained for the different widths of the fitting range. Colors denote the choices for the upper end of the fitting range for structure functions (sfc) and power spectra (psd); the lower end is the same as given in sec. 3. Different symbols denote the four experiments. The dashed black lines in the upper panels mark 4/3 and 1:1 proportion as in Figs. 4 and 5. In the lower panels, they mark the values of 2/3 and 5/3 as in Fig. 6. Note the axes limits are different than in Figs. 4-6.

*Author contributions.* JLN designed and performed the analysis. JLN, ML, DHL and SPM wrote the manuscript.

*Competing interests.* Szymon P. Malinowski is a member of the editorial board of Atmospheric Measurement Techniques.

*Acknowledgements.* We acknowledge the scientists and technical staff who contributed to the turbulence measurements in the four field experiments which are considered in this study. This material is based upon work supported by the NSF National Center for Atmospheric Research, which is a major facility sponsored by the National Science Foundation under Cooperative Agreement No. 1852977. JLN and SPM were funded by the European Union's Horizon 2020, within the project nextGEMS (grant no. 101003470). JLN was also supported by the Foundation for Polish Science (FNP).

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
