# Peer review of "The ratio of transverse to longitudinal turbulent velocity statistics for aircraft measurements"

_EGUsphere, 2024_

## Referee Comment (RC2)

**Review of "The ratio of transverse to longitudinal turbulent velocity statistics for aircraft measurements" by Jakub L. Nowak, Marie Lothon, Donald H. Lenschow, and Szymon P. Malinowski**

July 4, 2024

**General comments**

This manuscript discusses an inherently difficult problem, local isotropy, in relation to aircraft turbulence measurements. It calls attention to the fact that the predicted 4/3 ratio of transverse to longitudinal velocity component spectra and structure functions in the inertial subrange is not observed in many such flights. Because isotropy is related to the accepted values for the Kolmogorov constant(s) in one-dimensional spectra, this is an important issue when one wants to estimate the rate $\varepsilon$ of dissipation of turbulence kinetic energy.

A related issue of the slope of the power spectra and structure functions is also investigated, and a large scatter *around* the predicted exponents is found.

The manuscript raises awareness to the problem, but does not bring a solution or new insights. This does not prevent it from being timely and deserving of publication. My comments therefore should be taken by the authors not as obligatory changes that need to be made to the manuscript, but rather as an interested dialogue about a few facets of a very difficult question.

First, I would like to mention that balloon measurements appear to agree more closely to isotropy; see Siebert et al. [2006]. Secondly, numerical analyses may bring insightful results: Akinlabi et al. [2019] obtained results for the $P_T/P_L$ ratio larger than 4/3 from DNS (contrary to the current manuscript's results). The authors may find their discussion of physical causes of anisotropy in the ABL useful. Finally, LES of the flow around the sensor has produced some very useful results regarding flow distortion in the case of sonic anemometers: see Huq et al. [2017]; maybe something similar could be proposed as a future study regarding aircraft measurements?

**Specific comments**

Fitting of power laws in figures 1 and 2 may be a little deceiving. Compensated spectra often display a concave curve, rather than a flat (horizontal) plateau in the assumed range

of frequencies associated with the inertial subrange. Maybe you can discern further details about the departs from $-5/3$ and $2/3$ by plotting, for example, $k^{5/3}P(k)$ versus $k$? As an example, see the figure below from Akinlabi et al. [2019]:

[Figure]

Another rather interesting plot would be $P_w/P_u$ versus $k$. This would allow to detect if at least the ratio is increasing with $k$, which would be indicative that local isotropy is being approached at higher (unresolved) frequencies. As usual, care has to be taken regarding noise, aliasing, and other high-frequency effects. In this regard, see next comment.

l. 130–134    Is it possible that the coarser spatial resolution of aircraft measuremnts in comparison to helicopter and balloon measurements is part of the problem? For example, Siebert et al. [2006] found ratios closer to $4/3$ from sonic anemometer data. The onset of isotropy may be gradual across a perceived inertial subrange. See the figure below, from Kaimal et al. [1972]:

[Figure]

Figure 22.   Plot of the ratios of $w$ and $u$ spectral estimates showing approach to the 4/3 ratio required for isotropy.

l. 140    Please clarify: if your coordinate system $xyz$ is such that $x$ is the direction that the aircraft flies, then there is a mean wind (with respect to the Earth) that in general *will not* be in the direction of $x$. On land stations, it is customary to rotate the data so that the mean wind vector is $(\bar{u}, 0, 0)$, but you do not mention a similar procedure. Therefore, it appears that in the aircraft reference frame there will be a $\bar{v}$ and possibly a $\bar{w}$. How does that impact, if at all, your results? Is this irrelevant because the aircraft's speed is so much greater than the average wind speed with respect to the Earth?

l. 254    Can the authors discuss more at length how buoyancy and possibly other effects impact isotropy? Some interesting discussion (as a starting point) can again be found in Akinlabi et al. [2019].

**References**

Akinlabi, E. O., Wacławczyk, M., Mellado, J. P., and Malinowski, S. P. (2019). Estimating turbulence kinetic energy dissipation rates in the numerically simulated stratocumulus cloud-top mixing layer: Evaluation of different methods. *Journal of the Atmospheric Sciences*, 76(5):1471–1488.

Huq, S., De Roo, F., Foken, T., and Mauder, M. (2017). Evaluation of probe-induced flow distortion of campbell csat3 sonic anemometers by numerical simulation. *Boundary-Layer Meteorology*, 165:9–28.

Kaimal, J. C., Wyngaard, J. C., Izumi, Y., and Coté, O. R. (1972). Spectral characteristics of surface-layer turbulence. *Q J R Meteorol Soc*, 98:563–589.

Siebert, H., Lehman, K., and Wendisch, M. (2006). Observations of small-scale turbulence and energy dissipation rates in the cloudy boundary layer. *J Atmos Sci*, 63:1451–1466.

---

## Author Comment (AC1)

**Authors' Response to the Anonymous Referee #1**

Jakub L. Nowak, Marie Lothon, Donald H. Lenschow, Szymon P. Malinowski

We are grateful to the Referee #1 for the comments and suggestions on our manuscript. We respond to them in detail below. The original review is given in black, our answers in blue.

The manuscript describes the ratio of transverse to longitudinal structure functions and power spectra in the inertial subrange, as measured by aircraft, and how the derived ratios and scaling exponents deviate from the classical theory of homogeneous and isotropic turbulence. Derived ratios are closer to 3/4 than to the theoretical 4/3 value. The authors conclude that the derived ratios and exponents mainly depend on how the velocity components were measured on the aircraft compared to a minor influence of ABL regimes and across different field experiments.

**General remarks**

The manuscript provides a profound and insightful analysis of the spectra and structure functions, with a rigorous and detailed description of methods used to derive the ratios and scaling exponents, including uncertainty analysis and sensitivity to the fitting range of the spectra. I only have a few suggestions for improving clarity and better contextualizing the work within the existing literature.

We followed the suggestions, expanded the literature review in the introduction and extended the discussion on potential solutions.

**Specific comments**

1. It would be helpful to the reader if the abstract could clearly state the overall aim of the manuscript before outlining the methodological details. Is it to document the derived ratios and exponents and their deviation from theoretical values?

   We added a relevant sentence to the abstract. The goal is to document the statistics of the ratios and exponents derived from aircraft observations, quantify their departures from theoretical predictions and point out the differences among the aircraft.

2. I think the introduction could be re-structured to lead from a broad scope to the specific purpose of this paper. For example, the second paragraph is a rather technical explanation of turbulence measurements on aircraft, which I find hard to contextualize when reading for the first time.

We reordered the content in the introduction so that it starts with theoretical background, then describes the previous experimental studies and closes with the overview of our study.

3. The introduction (L. 63ff) mentions a few studies of isotropy from ground-based measurements. It would be interesting to mention what these studies find, since they exclude aircraft-specific errors. Also, are there studies available from tall towers (several hundreds of meters high), reaching above the surface layer?

We described the relevant findings from the ground-base measurements in the surface layer (Kaimal et al., 1972; Katul et al., 1995, 1997; Siebert and Muschinski, 2001; Chamecki and Dias, 2004). Moreover, to our literature review we added studies involving tethered balloon (Kaimal et al., 1972) and tall tower measurements (Kaimal et al., 1982). Those experiment reached above the surface layer into the mixed convective layer. Besides, we corrected the information given about the work by Siebert et al. (2006) who performed measurements in shallow cumulus clouds. In fact, their ACTOS platform was then carried by a tethered balloon, not a helicopter. A helicopter was indeed used for the same platform but in subsequent studies by the same group, e.g. in Nowak et al. (2021) whose results on anisotropy in the stratocumulus-topped ABL were also delineated in more detail in the revised introduction.

4. L. 72: Could you explain what the upwash distortion is? (Since it is mentioned several times.)

Upwash flow distortion appears forward of the aircraft due to the air being deflected above and below the wing when approaching an airfoil. We added a short explanation to the introduction. In the revised manuscript we used the term 'upstream flow distortion' instead for better comprehension.

5. L. 134: Could you still briefly mention TAS and sampling frequency for the ATR?

The TAS and sampling frequency for the ATR is the same as for the C130. We clarified this in the text.

6. It would be helpful to add an explanation about the applicability of the findings to the undisturbed ABL (and reference coordinate system along the mean wind), since the longitudinal and transversal components refer to the aircraft-referenced coordinate system (L. 136ff), including aircraft heading and pitch if I understand correctly.

We wrote an additional paragraph in the theoretical part of the introduction to explain how frozen flow approximation is applied and what directions are longitudinal/transverse in two typical experimental configurations: rapidly moving aircraft and fixed ground-based mast. Together with the clarified definition of the coordinate system in sec. 2.2., this should help readers to understand the conventions we use.

We are not entirely sure what is meant by undisturbed ABL. If the lack of detectable mean wind, then with a fixed point measurement (e.g. at a mast) one cannot obtain multi-point statistics like structure functions or power spectra. Actually, this is a case of extreme invalidity of Taylor's hypothesis because the turbulence intensity, i.e. the ratio of turbulence velocity scale to mean wind magnitude, becomes infinite. On the other hand, no mean wind does not pose a problem for aircraft measurements because the velocity of the aircraft with respect to air is relevant here.

7. L. 139: Add the reference coordinate system.

We specified the reference coordinate system by referring to an extra explanation of longitudinal/transverse directions given in the introduction.

8. L. 142: "Note that both vertical and lateral wind components are considered transverse." It might be helpful to mention this earlier, since both terms are used multiple times before.

It was mentioned in the extra paragraph about frozen flow approximation and longitudinal/transverse directions added to the revised introduction.

9. L. 210: Does the 3/4 value have a special meaning, or is it just by coincidence the closest value for the ratios?

No, it is coincidence. We removed the 3/4 lines from the plots.

10. Looking at Fig.3-Fig.5, one could interpret that filled symbols show less scatter than open circles. Could one conclusion of the study be that wind-parallel flight legs provide statistically more robust results than wind-perpendicular flight legs?

Discerning whether wind-parallel or wind-perpendicular flight legs provide statistically more robust results is not straight-forward. In general, it may depend on the dominant convection pattern as some of the structures are expected to align with the mean wind direction and some are not.

The specific difference in the apparent scatter in our Figs. 3-5 is, to a large extent, due to the different number of legs in each category. For example, during EUREC4A most legs were wind-perpendicular. Therefore, we added information about the numbers of wind-parallel and wind-perpendicular legs to Table 1. Please note that it already involves the change in segment classification according to levels for POST which we described at the end of this document in the section "Other changes".

We suppose that "statistical robustness" in the situation of different number of elements can be interpreted quantitatively as smaller standard error of the mean $s_{\overline{x}} = \sigma_x / \sqrt{N}$ where $\sigma_x$ is standard deviation and $N$ is the number of $x$ samples. The results on $s_{\overline{x}}$ given in the Table R1 below are ambiguous: depending on the variable and level the values are smaller either for the wind-parallel or wind-perpendicular segments.

11. L. 289ff: To investigate aircraft-specific errors leading to the deviation from theoretical values: What about a comparison to tall tower measurements or airborne Sonic anemometer measurements? Would the spatial distance between aircraft and reference measurement hinder a direct comparison?

We agree that a comparison to tall tower measurements will help to investigate aircraft-specific errors and eventually find physical explanations for our results. Similar strategy was applied by Kaimal et al. (1982) but over gently rolling terrain and with a different measurement technique used on the aircraft. Unfortunately, the tower of Boulder Atmospheric Observatory no longer exists. Ideally, a tower should be on a horizontally homogeneous surface and reasonably high to offer a meaningful comparison with aircraft. We added a paragraph proposing such an experiment to the discussion section.

**Table R1.** Standard error of the mean $s_{\overline{x}} = \sigma_x/\sqrt{N}$ for the variables given in the head row, separately for characteristic levels and flight segment orientation with respect to the mean wind.

| Level | Wind | N | $D_v/D_u$ | $P_v/P_u$ | $D_w/D_u$ | $P_w/P_u$ | $s_u$ | $s_v$ | $s_w$ | $p_u$ | $p_v$ | $p_w$ |
|---|---|---|---|---|---|---|---|---|---|---|---|---|
| ATR-EUREC4A | | | | | | | | | | | | |
| cloud-base | ⊥ | 116 | 0.008 | 0.007 | 0.010 | 0.011 | 0.004 | 0.006 | 0.006 | 0.011 | 0.011 | 0.008 |
| top-subcloud | ∥ | 11 | 0.008 | 0.009 | 0.022 | 0.027 | 0.009 | 0.009 | 0.018 | 0.036 | 0.032 | 0.019 |
| top-subcloud | ⊥ | 9 | 0.020 | 0.014 | 0.041 | 0.036 | 0.012 | 0.020 | 0.026 | 0.014 | 0.017 | 0.026 |
| mid-subcloud | ∥ | 9 | 0.020 | 0.018 | 0.026 | 0.021 | 0.016 | 0.008 | 0.010 | 0.018 | 0.017 | 0.016 |
| mid-subcloud | ⊥ | 10 | 0.016 | 0.015 | 0.024 | 0.012 | 0.010 | 0.011 | 0.020 | 0.016 | 0.018 | 0.019 |
| near-surface | ∥ | 5 | 0.010 | 0.015 | 0.028 | 0.023 | 0.018 | 0.027 | 0.025 | 0.022 | 0.045 | 0.031 |
| near-surface | ⊥ | 5 | 0.023 | 0.018 | 0.013 | 0.024 | 0.018 | 0.011 | 0.014 | 0.049 | 0.022 | 0.035 |
| TO-POST | | | | | | | | | | | | |
| cloud-top | ∥ | 9 | 0.058 | 0.080 | 0.097 | 0.127 | 0.015 | 0.056 | 0.036 | 0.017 | 0.055 | 0.055 |
| cloud-top | ⊥ | 32 | 0.025 | 0.031 | 0.039 | 0.051 | 0.009 | 0.016 | 0.018 | 0.015 | 0.021 | 0.021 |
| cloud-base | ∥ | 11 | 0.046 | 0.062 | 0.048 | 0.065 | 0.016 | 0.023 | 0.020 | 0.046 | 0.047 | 0.043 |
| cloud-base | ⊥ | 11 | 0.064 | 0.074 | 0.068 | 0.082 | 0.015 | 0.054 | 0.034 | 0.028 | 0.024 | 0.049 |
| sub-cloud | ∥ | 4 | 0.020 | 0.030 | 0.060 | 0.101 | 0.020 | 0.026 | 0.056 | 0.048 | 0.026 | 0.107 |
| sub-cloud | ⊥ | 38 | 0.013 | 0.014 | 0.020 | 0.021 | 0.009 | 0.010 | 0.010 | 0.019 | 0.010 | 0.015 |
| near-surface | ∥ | 4 | 0.011 | 0.018 | 0.029 | 0.013 | 0.046 | 0.045 | 0.028 | 0.056 | 0.109 | 0.189 |
| near-surface | ⊥ | 45 | 0.010 | 0.012 | 0.013 | 0.014 | 0.007 | 0.010 | 0.008 | 0.019 | 0.014 | 0.018 |

Concerning a comparison against airborne sonic anemometer, we expect it would be more difficult to discern the influence of platform-specific aerodynamic issues. A sonic anemometer was flown onboard ACTOS platform carried by a helicopter, e.g. in Nowak et al. (2021). They reported that $P_w/P_u$ strongly decreases already for $\lambda < 5$ m while the sonic path is 15 cm. The reason for such a behaviour at those wavelengths was not thoroughly explained.

12. Appendix C: Does the conclusion of appendix C mean that the insensitivity to the included scales extends beyond the inertial subrange, meaning that even the larger scales are close to anisotropy in the ABL? It would be interesting to see if measured anisotropy in the inertial subrange is related to anisotropy at larger scales. Were strongly anisotropic cases (at larger scales) considered in the analysis?

We did not investigate large-scale anisotropy and, in particular, did not condition the data on the properties related to larger scales. Therefore, strongly anisotropic cases at larger scales could be included in the analysis. We suppose the range of scales considered in Appendix C does not extend far beyond the inertial subrange if at all. The largest considered scales are 2.8 $L$, which for $L \sim 200$ m (c.f. Table 1) means $\sim 560$ m. This is typically smaller than the ABL depth $z_i$ while

Kaimal et al. (1976) and Kaimal et al. (1982) found that inside the mixed layer the peak of the power spectra is typically at $\lambda \sim 1.5 z_i$. Please also note that our particular method of computing $L$ is based on the e-decay of the autocorrelation function. It provides rather lower bound on the estimation of integral length scale as the e-crossing is always at smaller scale than the zero-crossing.

Following the suggestions by Referee #2, we expanded the analysis to scale-by-scale transverse-to-anisotropy ratios. Those additional results clearly indicate that anisotropy changes across scales. However, the largest of the considered scales seem to be closer to isotropy than the smallest ones, which we cannot explain with physical mechanisms relevant to the ABL.

In general, in contrast to the inertial subrange large scales are expected to exhibit anisotropy due to vertical confinement, imposed by the surface and stable inversion layer at ABL top, and due to inherent anisotropy of the turbulence generation mechanisms such as heat fluxes or wind shear. Convection in the ABL often organizes into low-aspect-ratio structures resembling cells, rolls etc. It would be indeed interesting to study whether and how the particular organization of convection affects the anisotropy of turbulence in the inertial subrange but, unfortunately, it is beyond the scope of our study.

**Technical comments**

1. The figure labels are very small and hard to read, especially Fig. 1, 2, B1 and C1. Would it be possible to plot fewer cases, but increase label sizes?

   We increased font size and restructured the figures. The number of panels in Figs. 1 and 2 was decreased to 4 and their size increased. The layout of Fig. B1 was changed from 3x4 into 4x3, which allowed for increasing the size of panels, and its style was improved to be more clear. In Fig. C1, we zoomed in to present relevant ranges.

2. L. 22: Maybe rather use "turbulence strength" than "turbulence intensity" to not confuse it with the actual variable TI.

   Sure. Corrected.

3. L. 59: boundaries of what?

   The boundaries of the domain for a turbulent fluid, e.g. sufficiently far from the surface and top of the ABL. Corrected.

**References**

Chamecki, M. and Dias, N. L.: The local isotropy hypothesis and the turbulent kinetic energy dissipation rate in the atmospheric surface layer, Quarterly Journal of the Royal Meteorological Society, 130, 2733–2752, https://doi.org/10.1256/QJ.03.155, 2004.

Kaimal, J. C., Wyngaard, J. C., Izumi, Y., and Coté, O. R.: Spectral characteristics of surface-layer turbulence, Quarterly Journal of the Royal Meteorological Society, 98, 563–589, https://doi.org/10.1002/QJ.49709841707, 1972.

Kaimal, J. C., Wyngaard, J. C., Haugen, D. A., Coté, O. R., Izumi, Y., Caughey, S. J., and Readings, C. J.: Turbulence Structure in the Convective Boundary Layer, Journal of Atmospheric Sciences, 33, 2152–2169, https://doi.org/10.1175/1520-0469(1976)033<2152:TSITCB>2.0.CO;2, 1976.

Kaimal, J. C., Eversole, R. A., Lenschow, D. H., Stankov, B. B., Kahn, P. H., and Businger, J. A.: Spectral Characteristics of the Convective Boundary Layer Over Uneven Terrain, Journal of Atmospheric Sciences, 39, 1098–1114, https://doi.org/10.1175/1520-0469(1982)039<1098:SCOTCB>2.0.CO;2, 1982.

Katul, G., Hsieh, C. I., and Sigmon, J.: Energy-inertial scale interactions for velocity and temperature in the unstable atmospheric surface layer, Boundary-Layer Meteorology, 82, 49–80, https://doi.org/10.1023/A:1000178707511, 1997.

Katul, G. G., Parlange, M. B., Albertson, J. D., and Chu, C. R.: Local isotropy and anisotropy in the sheared and heated atmospheric surface layer, Boundary-Layer Meteorology, 72, 123–148, https://doi.org/10.1007/BF00712392, 1995.

Nowak, J. L., Siebert, H., Szodry, K. E., and Malinowski, S. P.: Coupled and decoupled stratocumulus-topped boundary layers: Turbulence properties, Atmospheric Chemistry and Physics, 21, 10 965–10 991, https://doi.org/10.5194/ACP-21-10965-2021, 2021.

Siebert, H. and Muschinski, A.: Relevance of a tuning-fork effect for temperature measurements with the Gill solent HS ultra-sonic anemometer-thermometer, Journal of Atmospheric and Oceanic Technology, 18, 1367–1376, https://doi.org/10.1175/1520-0426(2001)018<1367:ROATFE>2.0.CO;2, 2001.

Siebert, H., Lehmann, K., and Wendisch, M.: Observations of small-scale turbulence and energy dissipation rates in the cloudy boundary layer, Journal of the Atmospheric Sciences, 63, 1451–1466, https://doi.org/10.1175/JAS3687.1, 2006.

---

## Author Comment (AC2)

**Authors' Response to the Anonymous Referee #2**

Jakub L. Nowak, Marie Lothon, Donald H. Lenschow, Szymon P. Malinowski

We are grateful to the Referee #2 for the comments and suggestions on our manuscript. We respond to them in detail below. The original review is given in black, our answers in blue.

**General comments**

This manuscript discusses an inherently difficult problem, local isotropy, in relation to aircraft turbulence measurements. It calls attention to the fact that the predicted 4/3 ratio of transverse to longitudinal velocity component spectra and structure functions in the inertial subrange is not observed in many such flights. Because isotropy is related to the accepted values for the Kolmogorov constant(s) in one-dimensional spectra, this is an important issue when one wants to estimate the rate $\varepsilon$ of dissipation of turbulence kinetic energy.

A related issue of the slope of the power spectra and structure functions is also investigated, and a large scatter around the predicted exponents is found.

The manuscript raises awareness to the problem, but does not bring a solution or new insights. This does not prevent it from being timely and deserving of publication. My comments therefore should be taken by the authors not as obligatory changes that need to be made to the manuscript, but rather as an interested dialogue about a few facets of a very difficult question.

First, I would like to mention that balloon measurements appear to agree more closely to isotropy; see Siebert et al. [2006].

True. We cited Siebert et al. (2006) in the introduction but mistakenly wrote that their ACTOS platform was carried by a helicopter instead of tethered balloon. A helicopter was indeed used for the same platform but in many subsequent studies by the same group, e.g. Nowak et al. (2021). In the revised introduction we reviewed in more details the tethered balloon measurements of Siebert et al. (2006) and Kaimal et al. (1976). See also our response to your specific comment below

Secondly, numerical analyses may bring insightful results: Akinlabi et al. [2019] obtained results for the $P_T/P_L$ ratio larger than 4/3 from DNS (contrary to the current manuscript's results). The authors may find their discussion of physical causes of anisotropy in the ABL useful.

Thanks for suggesting this study. We discussed the possible physical causes for turbulence anisotropy in sec. 5. However, we consider them unlikely to explain our results. See also our response to your specific comment below.

Finally, LES of the flow around the sensor has produced some very useful results regarding flow distortion in the case of sonic anemometers: see Hug et al. [2017]; maybe something similar could be proposed as a future study regarding aircraft measurements?

We agree that numerical modeling can be really useful in quantifying the influence of flow distortion on measurements; in particular in the situations where no laboratory or wind tunnel characterization is possible as is the case for aircraft fuselage. However, the velocities relevant for aircraft are much larger than mean wind in the ABL. Therefore, the common assumption of incompressibility is no longer valid and an adequate model needs to account for it. Definitely, simulations of the flow for the case of a 5-hole probe on the aircraft nose would be welcomed and interesting contribution but are beyond the scope of this study. We added a remark about potential benefits of numerical modeling to sec. 5.

**Specific comments**

Fitting of power laws in figures 1 and 2 may be a little deceiving. Compensated spectra often display a concave curve, rather than a flat (horizontal) plateau in the assumed range of frequencies associated with the inertial subrange. Maybe you can discern further details about the departs from —5/3 and 2/3 by plotting, for example, $k^{5/3}P(k)$ versus $k$? As an example, see the figure below from Akinlabi et al. [2019].

We plotted and inspected the compensated statistics as suggested, see them below for the example segments included in Figs. 1 and 2. They show our choice of fitting range is reasonable although there is non-zero slope throughout the range in some cases. Please note that this choice needs to be practical enough to involve sufficient number of data points in the fit and universal enough to be applicable to various segments. Also, for our analysis the fitting range needs to be the same for all three velocity components to obtain meaningful ratios in the next step.

In the revised manuscript, we decided to keep the non-compensated statistics in Figs. 1 and 2 so that in the very first figures of our paper the data is presented in a simple form common for experimental works. Nevertheless, we improved their style, drew reference 2/3 or -5/3 scalings and the extent of the fitting range. One compensated plot was used in a new Fig. 3, which is intended to illustrate the computations of scale-by-scale transverse-to-longitudinal ratios.

Worth to mention, Eqs. (5) and (6) involve only single free parameter and our method of fitting them to non-compensated data is numerically equivalent to taking the mean of compensated data within the selected range. Therefore, the choice between non-compensated and compensated does not affect those results.

Another rather interesting plot would be $P_w/P_u$ versus $k$. This would allow to detect if at least the ratio is increasing with $k$, which would be indicative that local isotropy is being approached at higher (unresolved) frequencies. As usual, care has to be taken regarding noise, aliasing, and other high-frequency effects. In this regard, see next comment.

We extended our analysis to consider the composite scale-by-scale transverse-to-longitudinal ratios. Please see the revised manuscript. However, the observed tendency is opposite, i.e. the ratios decrease and increasingly depart from 4/3 with decreas-

[Figure]

**Figure R1.** Compensated structure functions (open circles) and power spectra (filled circles) for a single segment from each experiment. The black dotted and dashed lines denote the chosen fitting ranges for structure functions and power spectra, respectively.

ing scale, in contrast to previous experimental studies on local isotropy and against the intuition that the anisotropy of large scales should be lost along the cascade towards small scales. See also our response to the comment below.

l. 130-134   Is it possible that the coarser spatial resolution of aircraft measurements in comparison to helicopter and balloon measurements is part of the problem? For example, Siebert et al. [2006] found ratios closer to 4/3 from sonic anemometer data. The onset of isotropy may be gradual across a perceived inertial subrange. See the figure below, from Kaimal et al. [1972].

It is very reasonable intuition that local isotropy should appear gradually with decreasing scale as showed by Kaimal et al. (1972) for the surface layer. Siebert et al. (2006) calculated $P_v/P_u$ and $P_w/P_u$ in the range of scales about 0.4–8 m. Actually. they analyzed two measurement series collected in shallow cumulus clouds. In the first one, both spectral ratios were approximately 4/3. In the second, also relatively close to the isotropic value, however with $P_v/P_u$ systematically higher and $P_w/P_u$ systematically lower than 4/3. Due to larger TAS of aircraft, we investigated larger scales: for power

spectra from $4\Delta r \approx 16$ m in the case of ATR and C130, and from $6\Delta r \approx 8.4$ m in the case of TO. Hence, our range is disjoint from the range of Siebert et al. (2006).

On the other hand, Nowak et al. (2021) analyzed measurements from the same platform ACTOS, however carried by a helicopter, in coupled and decoupled marine stratocumulus-topped ABLs during ACORES campaign (Siebert et al., 2021). They showed that $P_w/P_u$ in the coupled ABL does reach 4/3 for the scales 5–100 m at a few sampled levels. Such scales should be at least partly resolved by research aircraft. In our revised introduction, we reviewed in more detail the findings on scale-by-scale spectral ratios from the studies mentioned above (Kaimal et al., 1972; Siebert et al., 2006; Nowak et al., 2021).

Following the suggestions, we computed the composite scale-by-scale transverse-to-longitudinal ratios for our data. In order to average over various segments in a meaningful way, we considered the non-dimensional scales $r/L$ and $\lambda/L$ for structure functions and power-spectra, respectively. The results exhibit no tendency to approach 4/3 with decreasing scale. Actually, the ratios decrease and increasingly depart from 4/3 with decreasing scale. This trend is consistent with the absolute values of the fitted scaling exponents which are larger for transverse than longitudinal velocity components.

Moreover, similarly to previously presented bulk parameters, the scale-by-scale ratios seem to be to a large extent aircraft specific as the differences between the characteristic levels of the ABL are not substantial. The exact reason for the discrepancy in the observed turbulence isotropy between the measurements performed with research aircraft and other platforms remains uncertain. Aircraft-dependence in our results suggest deficiencies of the measurement technique. The goal of our study is in fact to raise awareness of this issue as well as stimulate further work to explain the discrepancies and potentially improve measurements.

l. 140 Please clarify: if your coordinate system $xyz$ is such that $x$ is the direction that the aircraft flies, then there is a mean wind (with respect to the Earth) that in general will not be in the direction of $x$. On land stations, it is customary to rotate the data so that the mean wind vector is $(\overline{u}, 0, 0)$, but you do not mention a similar procedure. Therefore, it appears that in the aircraft reference frame there will be a $\overline{v}$ and possibly a $\overline{w}$. How does that impact, if at all, your results? Is this irrelevant because the aircraft's speed is so much greater than the average wind speed with respect to the Earth?

We wrote an additional paragraph in the theoretical part of the introduction to explain how frozen flow approximation is applied and which directions are longitudinal/transverse in two typical experimental configurations: rapidly moving aircraft and fixed ground-based mast.

Our coordinate system $xyz$ is such that $x$ is the direction in which the aircraft moves with respect to air. For stabilized segments it coincides with the orientation of the aircraft fuselage projected onto horizontal plane (true heading angle). $x$ is not the direction in which the aircraft moves with respect to Earth (course over ground). The air velocity with respect to Earth (i.e. wind) and the aircraft motion with respect to Earth are irrelevant for the definition of the coordinate system. Mean wind vector expressed in our coordinate system is generally $(\overline{u}, \overline{v}, 0)$, i.e. has non-zero longitudinal and lateral components. The measurement records are anyway detrended before calculating structure functions or power spectra, therefore we do not expect the values of $\overline{u}$ and $\overline{v}$ to influence the results.

l. 254 Can the authors discuss more at length how buoyancy and possibly other effects impact isotropy? Some interesting discussion (as a starting point) can again be found in Akinlabi et al. [2019].

We described the potential influence of buoyancy and wind shear on isotropy in sec. 5. However, although buoyancy and wind shear certainly affect the character of turbulence, we believe it is unlikely these factors explain our results. It is because $D_w/D_u$, $P_w/P_u$ are smaller than 4/3 at all levels of the ABL (see Table 2) and almost all considered scales. Even if there was very strong wind shear, it should be restricted to the surface and the top of the ABL. In the middle, one would then need stable stratification to weaken $D_w$, $P_w$ but it is not the case.

**References**

Kaimal, J. C., Wyngaard, J. C., Izumi, Y., and Coté, O. R.: Spectral characteristics of surface-layer turbulence, Quarterly Journal of the Royal Meteorological Society, 98, 563–589, https://doi.org/10.1002/QJ.49709841707, 1972.

Kaimal, J. C., Wyngaard, J. C., Haugen, D. A., Coté, O. R., Izumi, Y., Caughey, S. J., and Readings, C. J.: Turbulence Structure in the Convective Boundary Layer, Journal of Atmospheric Sciences, 33, 2152–2169, https://doi.org/10.1175/1520-0469(1976)033<2152:TSITCB>2.0.CO;2, 1976.

Nowak, J. L., Siebert, H., Szodry, K. E., and Malinowski, S. P.: Coupled and decoupled stratocumulus-topped boundary layers: Turbulence properties, Atmospheric Chemistry and Physics, 21, 10 965–10 991, https://doi.org/10.5194/ACP-21-10965-2021, 2021.

Siebert, H., Lehmann, K., and Wendisch, M.: Observations of small-scale turbulence and energy dissipation rates in the cloudy boundary layer, Journal of the Atmospheric Sciences, 63, 1451–1466, https://doi.org/10.1175/JAS3687.1, 2006.

Siebert, H., Szodry, K.-E., Egerer, U., Wehner, B., Henning, S., Chevalier, K., Lückerath, J., Welz, O., Weinhold, K., Lauermann, F., Gottschalk, M., Ehrlich, A., Wendisch, M., Fialho, P., Roberts, G., Allwayin, N., Schum, S., Shaw, R. A., Mazzoleni, C., Mazzoleni, L., Nowak, J. L., Malinowski, S. P., Karpinska, K., Kumala, W., Czyzewska, D., Luke, E. P., Kollias, P., Wood, R., and Mellado, J. P.: Observations of Aerosol, Cloud, Turbulence, and Radiation Properties at the Top of the Marine Boundary Layer over the Eastern North Atlantic Ocean: The ACORES Campaign, Bulletin of the American Meteorological Society, 102, E123–E147, https://doi.org/10.1175/bams-d-19-0191.1, 2021.